# Inductive Domain Transfer In Misspecified Simulation-Based Inference

**Ortal Senouf** [*]
EPFL
Lausanne, Switzerland

**Antoine Wehenkel**
Apple
Zürich, Switzerland

**Cédric Vincent-Cuaz**
EPFL
Lausanne, Switzerland

**Emmanuel Abbé**
EPFL, Apple
Lausanne, Switzerland

**Pascal Frossard**
EPFL
Lausanne, Switzerland

## Abstract

Simulation-based inference (SBI) of latent parameters in physical systems is often hindered by model misspecification–the mismatch between simulated and real-world observations caused by inherent modeling simplifications. RoPE, a recent SBI approach, addresses this challenge through a two-stage domain transfer process that combines semi-supervised calibration with optimal transport (OT)-based distribution alignment. However, RoPE operates in a fully transductive setting, requiring access to a batch of test samples at inference time, which limits scalability and generalization. We propose a fully inductive and amortized SBI framework that integrates calibration and distributional alignment into a single, end-to-end trainable model called FRISBI. Our method leverages mini-batch OT with a closed-form coupling to align real and simulated observations that correspond to the same latent parameters, using both paired calibration data and unpaired samples. A conditional normalizing flow is then trained to approximate the OT-induced posterior, enabling efficient inference without simulation access at test time. Across a range of synthetic and real-world benchmarks–including complex medical biomarker estimation–our approach matches or exceeds the performance of RoPE, while offering improved scalability and applicability in challenging, misspecified environments.

## 1   Introduction

Inference of latent variables that describe important properties of physical systems is a fundamental problem in many domains, including environmental [1, 2], mechanical [3, 4, 5], and physiological [6, 7, 8] systems. Traditionally, this problem has been approached by formulating a mathematical model that relates the observations $x$ to the latent parameters of interest $\theta$, and solving the corresponding inverse problem to infer $\theta$ from $x$ [9, 10].

Modern machine learning (ML) has achieved remarkable success in complex tasks, sparking interest in its application to inferring latent parameters from observations. However, standard supervised learning is often impractical in this context, as ground truth parameter data is typically expensive or infeasible to obtain, such as in medical applications where direct measurement may require invasive procedures. To address this, two prominent approaches have emerged: *simulation-based inference* (SBI) [11] and *hybrid learning* [12, 13, 14]. SBI trains ML models on simulated data to directly estimate parameters while capturing uncertainty through posterior estimation, becoming a

---

[*]Corresponding Author, `ortal.senouf@epfl.ch`

39th Conference on Neural Information Processing Systems (NeurIPS 2025).

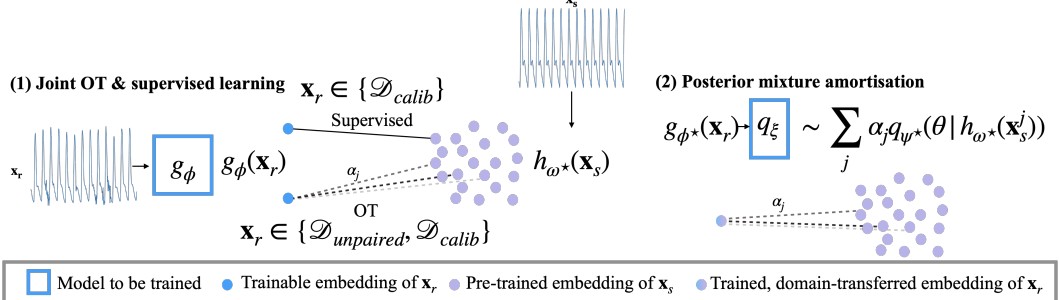

Figure 1: **FRISBI Overview.** Similar to RoPE [22], we assume a trained *neural statistics encoder* (NSE), $h_{\omega^\star}$, that maps simulation data $\boldsymbol{x}_s$ to embeddings $h_{\omega^\star}(\boldsymbol{x}_s)$, and a *neural posterior estimator* (NPE), $q_{\psi^\star}$, which estimates simulated posterior distributions. FRISBI performs: **(1)** Joint optimal transport (OT) and supervised learning. Both paired and unpaired samples contribute to the OT plan (dashed lines), weighted by $\alpha_j$. Supervised samples from $\mathcal{D}_{calib}$ (solid lines) anchor the OT matching. The real-observations encoder $g_\phi$ is fine-tuned to optimize representations for both supervised learning and OT-based domain transfer. **(2)** A *conditional density estimator*, $q_\xi$, approximates the posterior arising from the OT-based mixture of posteriors.

cornerstone in scientific domains [15, 16, 17, 18, 19], though it can suffer from sensitivity to model misspecification [20, 21]. In hybrid learning, the simulator is integrated with ML components to obtain a more accurate model of the system. While this approach provides a useful inductive bias, it often requires simulators that are differentiable and computationally feasible, limiting its applicability to realistic and complex systems.

A recent work introduces RoPE [22], an SBI approach designed to address model misspecifications through a two-stage, semi-supervised domain transfer strategy. The first stage focuses on pointwise calibration using a small set of labeled real observations—i.e., observations for which the corresponding parameters $\boldsymbol{\theta}$ are known, whereas the second stage aligns distributions using optimal transport (OT). While effective, this approach is inherently transductive, it requires a batch of test samples for inference, limiting its applicability in scenarios that require inductive inference. In many real-world settings, access to a batch of test-time observation is unrealistic, and the inferred posterior for a given single test input can vary depending on the batch it is embedded in—undermining stability and reproducibility. Additionally, the strict separation between pointwise and distribution-wise alignment may prevent full exploitation of their complementary strengths.

In this work, we propose a new framework, illustrated in Fig. 1, that builds upon elements of RoPE by amortizing the optimal transport (OT) step through a mini-batch unbalanced OT approach [23, 24]. Similarly to RoPE, our method assumes access to a limited set of ground-truth pairs of observations $\boldsymbol{x}$ and parameters $\boldsymbol{\theta}$. It features a closed-form solution for the transport plan and offers two key advantages:

- **Inductive Joint Training of Alignment Steps:** Enables end-to-end training of both pointwise and distribution-wise alignment, better leveraging their complementary strengths.

- **Amortised Posterior Estimation:** Provides a scalable, inductive solution for OT-based posterior estimation, eliminating the need to repeatedly access simualtions during inference.

We show that our method, while fully inductive and applicable to individual test samples, achieves competitive—and often superior—performance compared to the transductive RoPE baseline across a range of benchmarks. This includes both synthetic and real-world datasets, with strong results in terms of accuracy and calibration, even in challenging settings such as complex biomarker estimation.

## 2 Background

### 2.1 Simulation-Based Inference and Neural Posterior Estimation

We consider a simulator $S : \boldsymbol{\theta} \longrightarrow \mathcal{X}$ that, given parameters $\boldsymbol{\theta} \sim p(\boldsymbol{\theta})$, produces simulated data $\boldsymbol{x}_s = S(\boldsymbol{\theta})$, whose likelihood $p(\boldsymbol{x}_s \mid \boldsymbol{\theta})$ is intractable. Simulation-based inference (SBI) methods

sidestep likelihood evaluation by training a neural conditional density estimator $q_\psi(\boldsymbol{\theta} \mid \boldsymbol{x}_s)$ (e.g. a conditional normalizing flow [25]) to approximate the posterior $p(\boldsymbol{\theta} \mid \boldsymbol{x}_s)$. Very often, when $\boldsymbol{x}_s$ is high-dimensional, a *neural statistics encoder* (NSE [22]) $h_\omega$ is used to obtain a lower-dimensional representation with sufficient information for the inference task. Eventually, in *neural posterior estimation* (NPE), one minimizes the expected negative log-likelihood over simulator draws w.r.t parameters $\boldsymbol{\theta}$ :

$$\mathcal{L}_{\mathrm{NPE}}(\psi, \omega) = \mathbb{E}_{\boldsymbol{\theta} \sim p(\boldsymbol{\theta}), \, \boldsymbol{x}_s \sim p(\cdot \mid \boldsymbol{\theta})} \big[ -\log q_\psi(\boldsymbol{\theta} \mid h_\omega(\boldsymbol{x}_s)) \big].$$

Under sufficient expressiveness of the density estimation model $q_\psi$ and the encoder $h_\omega$, while considering access to arbitrarily large simulated data $\mathcal{D}_{\mathrm{SBI}} = \{(\boldsymbol{\theta}_j, \boldsymbol{x}_s^j)\}_{j=1}^{N_{\mathrm{SBI}}}$, the density $p(\boldsymbol{\theta} \mid \boldsymbol{x}_s)$ can be approximated by sampling from $q_{\psi^\star}(\boldsymbol{\theta} \mid h_{\omega^\star}(\boldsymbol{x}_s))$ [26][27].

## 2.2 Transductive Semi-Supervised Posterior Estimation (RoPE)

In the presence of model misspecification, the simulator-induced posterior $p(\boldsymbol{\theta} \mid \boldsymbol{x}_s)$ may be biased relative to the true real-world posterior $p(\boldsymbol{\theta} \mid \boldsymbol{x}_r)$. Consequently, both NSE $h_{\omega^\star}$ and the NPE $q_{\psi^\star}$, when trained solely on simulated data, may fail to perform reliably on real observations. RoPE [22], on which this work builds, addresses this in two stages.

**NSE Fine-tuning.** Once the NSE, $h_{\omega^\star}$, and NPE, $q_{\psi^\star}$ from Section 2.1 are trained and fixed, a small set (*calibration set*) of real observations $\boldsymbol{x}_r^i$ and their corresponding known parameters $\boldsymbol{\theta}_i$, is used to adapt the encoder to the domain shift. Each $\boldsymbol{\theta}_i$ is passed through the simulator to obtain the corresponding $\boldsymbol{x}_s^i = S(\boldsymbol{\theta}_i)$ and the calibration set becomes $\mathcal{D}_{calib} = \{\boldsymbol{x}_r^i, \boldsymbol{x}_s^i\}_{i=1}^{N_{calib}}$. Then, $g_\phi$, the NSE for the real observations, initialized as $h_{\omega^\star}$, is fine-tuned on $\mathcal{D}_{calib}$ to minimize the mean squared error between $g_\phi(\boldsymbol{x}_r^i)$ and $h_{\omega^\star}(\boldsymbol{x}_s^i)$ for every pair of $\boldsymbol{x}_r^i, \boldsymbol{x}_s^i \in \mathcal{D}_{calib}$. The outcome is a limited (depending on the size of $\mathcal{D}_{calib}$) domain adaptation of the NSE $g_\phi$, enabling it to encode representations of real observations in the same latent space as $h_{\omega^\star}(\mathbf{x}_s)$.

**Entropic OT coupling.** Since NSE fine-tuning relies on a limited calibration set, its ability to generalize to unseen real observations is constrained, leaving residual uncertainty in the domain transfer. RoPE takes this uncertainty into account by coupling real and simulated observations through entropic OT, which computes a soft assignment matrix between embeddings $\{g_{\phi^\star}(\boldsymbol{x}_r^i)\}$ of test observations $\mathcal{D}_{test} = \{\boldsymbol{x}_r^i\}_{i=1}^{N_{test}}$ and embeddings $\{h_{\omega^\star}(\boldsymbol{x}_s^j)\}$ of a fresh simulation set $\mathcal{D}_{OT} = \{\boldsymbol{x}_s^j\}_{j=1}^{N_{OT}}$. The latter then acts as prototypes to which the matched embeddings $\{g_{\phi^\star}(\boldsymbol{x}_r^i)\}$ must be close in an Euclidean sense, similarly to the soft Kmeans algorithm [28]. Specifically, they propose to solve for the following semi-balanced entropic OT problem [29, 30]:

$$\boldsymbol{P}^\star = \arg \min_{\boldsymbol{P} \in \mathcal{B}(N_{test}, N_{OT})} \langle \boldsymbol{P}, \boldsymbol{C} \rangle + \rho \, \mathrm{KL}\big(\boldsymbol{P}^\top \mathbf{1}_{N_{test}} \, \| \, \tfrac{1}{N_{OT}} \mathbf{1}_{N_{OT}}\big) + \gamma \langle \boldsymbol{P}, \log \boldsymbol{P} \rangle. \quad (1)$$

where $\boldsymbol{P}$ is constrained row-wise to $\mathcal{B}(N_{test}, N_{OT}) = \{\boldsymbol{P} \in \mathbb{R}_+^{N_{test} \times N_{OT}} | \boldsymbol{P}\mathbf{1}_{N_{OT}} = \tfrac{1}{N_{test}} \mathbf{1}_{N_{test}}\}$ and $\boldsymbol{C}$ is the pairwise euclidean distance matrix between both sets of embeddings. The weight $\rho$ encourages prototypes to be matched to a uniform number of test samples, enabling unbalanced OT to accommodate prior misspecification, while $\gamma$ controls the entropy regularization strength, thereby tuning the method's sensitivity to model misspecification and to uncertainty introduced during encoder fine-tuning on the calibration set. Problem 1 is commonly solved using an iterative bregman projection solver [31, 32, 33]. As detailed in Appendix, it comes down to actualize along iterations $t$ the transport plan following:

$$\boldsymbol{P}^{(t+1)} \leftarrow \mathrm{diag}\big(\tfrac{\mathbf{1}_{N_{test}}}{N_{test} \boldsymbol{K}^{(t)} \mathbf{1}}\big) \boldsymbol{K}^{(t)} \quad \text{with} \quad \boldsymbol{K}^{(t)} = \exp\left(\frac{-\boldsymbol{C} - \rho \mathbf{1} \log(N_{OT} \boldsymbol{P}^{(t)\top} \mathbf{1})^\top}{\gamma}\right) \quad (2)$$

Finally, [22] shows that the calibrated posterior for each $\boldsymbol{x}_r^i$ can be approximated by marginalizing over $\boldsymbol{x}_s$, resulting in the following posterior:

$$\tilde{p}(\boldsymbol{\theta} \mid \boldsymbol{x}_r^i) := \sum_{j=1}^{N_{OT}} \alpha_{ij} \, q_{\psi^\star}(\boldsymbol{\theta} \mid h_{\omega^\star}(\boldsymbol{x}_s^j)) \quad \text{with} \quad \alpha_{ij} = N_{test} P_{ij}^\star. \quad (3)$$

# 3 Methods

RoPE, while offering robust posterior estimation, requires computing the OT coupling over the entire test batch $\mathcal{D}_{test}$ at once, rendering the approach inherently **transductive**. This limits its ability to generalize to unseen observations without re-computing the transport plan.

We propose a new Framework for Robust Inductive domain transfer in misspecified Simulation-Based Inference named FRISBI. It relies on a unified workflow that achieves a joint distribution-level and point-wise alignment while enabling **inductive** inference, thereby extending the solution to misspecified SBI beyond the limitations described above. The encoder $g_\phi$ is first trained using a joint objective that combines a variant of entropic OT admitting closed-form solutions, with a supervised calibration loss, as detailed in Section 3.1. Then to avoid the need for accessing simulations at test time, we further amortize this solution using the inductive strategy presented in Section 3.2. A complete description of the full pipeline and training procedure is provided in Algorithm 3.2. For clarity, Appendix A includes a summary table describing the different datasets.

## 3.1 Balancing Unpaired Alignment and Point-wise Domain Transfer

In addition to the data assumed to be available in RoPE [22], we also assume access to a large, **unpaired** dataset of real observations, denoted as $\mathcal{D}_u = \{\boldsymbol{x}_r^i\}_{i=1}^{N_u}$. Furthermore, we can generate a large set of simulations, separated from the one used to train the NPE, forming the dataset $\mathcal{D}_{OT} = \{\boldsymbol{x}_s^j\}_{j=1}^{N_{OT}}$. We propose to learn an encoder $g_\phi$ that optimizes a joint objective, denoted $\mathcal{L}_{\text{joint}}$, composed of two terms. The first component coincides with the entropic OT objective used in RoPE (see Eq. (1)), with the column-marginal constraint parameter fixed at $\rho = 0$. It defines a coupling between the encoded real samples $g_\phi(\boldsymbol{x}_r^i)$ and the fixed simulated representations $h_{\omega^\star}(\boldsymbol{x}_s^j)$. The second component operates on the calibration set $\mathcal{D}_{\text{calib}}$ and aims to control the deviation of the embeddings of real samples from those of their paired simulated samples. Formally, $g_\phi$ is trained by solving the following problem:

$$\arg\min_\phi \underbrace{\sum_{\substack{\boldsymbol{x}_r \sim D_r \\ \boldsymbol{x}_s \in D_s}} \left[ P_{ij} \| g_\phi(\boldsymbol{x}_r^i) - h_{\omega^\star}(\boldsymbol{x}_s^j) \|^2 + \gamma\, P_{ij} \log P_{ij} \right]}_{\text{Entropic OT}} + \lambda \underbrace{\sum_{\substack{\boldsymbol{x}_r, \boldsymbol{x}_s \\ \in D_{calib}}} \| g_\phi(\boldsymbol{x}_r^i) - h_{\omega^\star}(\boldsymbol{x}_s^i) \|^2}_{\text{Supervised Loss}}, \quad (4)$$

where $\boldsymbol{P} \in \mathcal{B}(N_u, N_{\mathbf{OT}})$ is an optimal coupling between both distributions and $(\gamma, \lambda)$ are regularization hyperparameters. We stress that for the supervised loss, all paired samples $(\boldsymbol{x}_r, \boldsymbol{x}_s)$ in the calibration set $\mathcal{D}_{calib}$ are taken. Whereas for the OT loss, the set $\mathcal{D}_r$ is a combination of a batch $\mathcal{B}_t$ sampled from the unpaired dataset $\mathcal{D}_u$ and samples from the calibration set $\mathcal{D}_{calib}$, while the set $\mathcal{D}_s$ consists of the entire simulation dataset $\mathcal{D}_{OT}$ as well as simulated samples $(\boldsymbol{x}_s)$ from the calibration set $\mathcal{D}_{calib}$. Intuitively, when calibration pairs are accurate (i.e., they share the exact same latent parameters $\boldsymbol{\theta}$), minimizing the supervised loss on the calibration set tends to sharpen the transport plan, resulting in alignment between the OT and supervised objectives. In contrast, when calibration pairs are noisy or mismatched, the two objectives may conflict, allowing the OT term to compensate for uncertainties in the calibration set.

We specifically enforce $\rho = 0$ in the entropic OT objective as the resulting optimization problem w.r.t $\boldsymbol{P}$ is naturally well-suited for inductive learning. Indeed, one can see that setting $\rho = 0$ in Eq. (1), implies that this problem admits a closed-form solution $\boldsymbol{P}^\star = \text{diag}(\frac{1_{N_u}}{N_u \boldsymbol{K} \mathbf{1}})\boldsymbol{K}$ where $\boldsymbol{K} = e^{-\boldsymbol{C}/\gamma}$ and $\boldsymbol{C}$ is the pairwise euclidean distance matrix between embeddings (see also [31, Proposition 1]). This leads to the efficient stochastic gradient descent (SGD) algorithm described in Stage 1 of Algorithm 3.2, which alternates between computing embeddings for real observations and independently computing the corresponding closed-form couplings for each embedding. Setting $\rho = 0$ enables this closed-form computation, requiring $N_{\text{OT}}$ operations per sample in $D_u$, each involving a Euclidean distance in $\mathbb{R}^d$, for an overall complexity of $\mathcal{O}(d)$. In the mini-batch setting, this yields a per-step complexity of $B_t N_{\text{OT}} d$, where $B_t$ is the batch size. Remark that an analogous strategy could be applied when $\rho > 0$, replacing the closed-form computation with iterative updates of Eq. 2 until convergence, as typically done in mini-batch OT [23, 24]. However, this approach is more computationally demanding and prone to bias, with high sensitivity to batch size. In addition,

RoPE, which uses $\rho > 0$ and cannot operate on mini-batches, scales as $\mathcal{O}(\log(N_u N_{\mathrm{OT}})N_u N_{\mathrm{OT}}d)$, where $N_u$ is the total number of real observations. These considerations further motivate our choice of $\rho = 0$ for improved efficiency and scalability.

Finally, once the model is trained, the posterior mixture coefficients $\alpha_{ij} = N_u P^\star_{ij}$ for a new test sample $\boldsymbol{x}^{test}_r$ can be directly computed by evaluating $C_{test,j}$ using the trained encoder $g_{\phi^\star}$ and the transport plan $\boldsymbol{P}^\star$, obtained in closed form. This yields the posterior mixture as defined in RoPE (Eq. 3).

## 3.2 Amortization of OT-based Posterior Estimation

Although the pipeline described in Section 3.1 enables inductive posterior estimation, it still relies on access to the same simulations used to train the loss in Eq. 4, at test time. To mitigate that, we propose to fit a conditional normalizing flow (cNF) $q_\xi$ that approximates the OT-based posterior mixture, conditioned directly on the real observation embeddings $g_{\phi^\star}(\boldsymbol{x}_r)$.

To fit $q_\xi$, we maximize its expected log density under the target mixture,

$$\arg\max_\xi \; \mathbb{E}_{\boldsymbol{\theta} \sim p_{\mathrm{target}}(\boldsymbol{\theta}|\boldsymbol{z})}\big[\log q_\xi(\boldsymbol{\theta} \mid \boldsymbol{z})\big].$$

where $p_{target}(\boldsymbol{\theta}|\boldsymbol{z})$ is computed as in Eq. 3. By the linearity of expectation, this is equivalent to

$$\sum_{j=1}^{N} \alpha_{ij}\, \mathbb{E}_{\boldsymbol{\theta} \sim q_{\psi^\star}(\boldsymbol{\theta}|h_{\omega^\star}(\boldsymbol{x}_s^j))}\big[\log q_\xi(\boldsymbol{\theta} \mid \boldsymbol{z})\big].$$

In practice, we approximate each inner expectation by drawing $K$ samples $\{\boldsymbol{\theta}^{(j,k)}\}_{k=1}^K$ from $q_{\psi^\star}(\boldsymbol{\theta} \mid h_{\omega^\star}(\boldsymbol{x}_s^j))$, and train $q_\xi$ with the set of unpaired real observations $\mathcal{D}_u$, and simulations $\mathcal{D}_{OT}$ on $\mathcal{L}_{flow}$:

$$\arg\min_\xi \; -\frac{1}{|\mathcal{B}_t|} \sum_{i \in \mathcal{B}_t} \Big[\frac{1}{K} \sum_{j=1}^{N_{OT}} \alpha_{ij} \sum_{k=1}^{K} \log q_\xi\big(\boldsymbol{\theta}^{(j,k)} \mid \boldsymbol{z}\big)\Big]. \tag{5}$$

During inference, for a given test embedding $\boldsymbol{z} = g_{\phi^\star}(\boldsymbol{x}_r^{test})$, $p(\boldsymbol{\theta} \mid \boldsymbol{x}_r)$ can be approximated by sampling directly from $q_{\xi^\star}(\boldsymbol{\theta} \mid \boldsymbol{z})$ without requiring any access to simulations.

---

**Training Procedure**

**Datasets:** $\mathcal{D}_u = \{\boldsymbol{x}_r^i\}_{i=1}^{N_u}, \quad \mathcal{D}_{OT} = \{\boldsymbol{x}_s^j\}_{j=1}^{N_{OT}}, \quad \mathcal{D}_{calib} = \{(\boldsymbol{x}_r^i, \boldsymbol{x}_s^i)\}_{i=1}^{N_{calib}}$

**Trained Models:** NSE $h_{\omega^\star}$, NPE $q_{\psi^\star}$

**Stage 1: Joint supervised & OT Training**

1: $\boldsymbol{w}_j \leftarrow h_{\omega^\star}(\boldsymbol{x}_s^j) \; \forall \boldsymbol{x}_s^j \in \mathcal{D}_{\mathcal{OT}}$
2: $\boldsymbol{w}_{ic} \leftarrow h_{\omega^\star}(\boldsymbol{x}_s^i) \; \forall \boldsymbol{x}_s^i \in \mathcal{D}_{calib}$
3: **for** $e = 1$ to epochs **do**
4:     **for** batch $\mathcal{B}_t : \{\boldsymbol{x}_r^i\}_{i \in \mathcal{B}_t}$ from $\mathcal{D}_u$ **do**
5:         $\boldsymbol{z}_i \leftarrow g_\phi(\boldsymbol{x}_r^i) \; \forall i \in \mathcal{B}_t$
6:         $\boldsymbol{z}_{ic} \leftarrow g_\phi(\boldsymbol{x}_r^i), \; \forall i \in \mathcal{D}_{calib}$
7:         $P_{ij} = \frac{1}{|\mathcal{B}_t|} \frac{\exp(-\|\boldsymbol{z}_i - \boldsymbol{w}_j\|^2/\gamma)}{\sum_j \exp(-\|\boldsymbol{z}_i - \boldsymbol{w}_j\|^2/\gamma)}$
8:         Compute $\mathcal{L}_{joint}$ 4 $\forall i \in \mathcal{B}_t, ic, j$
9:         Update $\phi$ by gradient step

**Stage 2: Conditional NF Amortization**

1: $\mathcal{Z} : \boldsymbol{z}_i \leftarrow g_{\phi^\star}(\boldsymbol{x}_r^i) \; \forall \boldsymbol{x}_r^i \in \mathcal{D}_u$
2: $\alpha_{ij} = \frac{\exp(-\|\boldsymbol{z}_i - \boldsymbol{w}_j\|^2/\gamma)}{\sum_j \exp(-\|\boldsymbol{z}_i - \boldsymbol{w}_j\|^2/\gamma)} \; \forall \boldsymbol{z}_i, \boldsymbol{w}_j$
3: **for** $e = 1$ to epochs **do**
4:     **for** batch $\mathcal{B}_t : \{\boldsymbol{z}_i\}_{i \in \mathcal{B}_t}$ from $\mathcal{Z}$ **do**
5:         Sample $\boldsymbol{\theta}_{j,k} \sim q_{\psi^\star}(\boldsymbol{\theta} \mid \boldsymbol{w}_j)$
6:         Compute $\mathcal{L}_{flow}$ 5, $\forall i \in \mathcal{B}_t, j, k$
7:         Update $\xi$ by gradient step

**Inference:** $\boldsymbol{z} = g_{\phi^\star}(\boldsymbol{x}_r), \quad p(\boldsymbol{\theta} \mid \boldsymbol{x}_r) \approx q_{\xi^\star}(\boldsymbol{\theta} \mid \boldsymbol{z})$    by sampling $\boldsymbol{\theta} \sim q_{\xi^\star}(\boldsymbol{\theta} \mid \boldsymbol{z})$

# 4 Experiments

## 4.1 Benchmarks

We evaluated our proposed approach on four benchmarks: a synthetic one, two real but controlled ones, and one complex real-world benchmark. In the synthetic setting, real observations are emulated using a more complex simulator. The two controlled benchmarks involve data sampled from real systems with experimental control. The final benchmark contains real-world observations collected "in the wild," with no control over sample distributions or nuisance variable variations. The first three benchmarks were also used to evaluate RoPE [22], the main baseline method.

**Pendulum.** A widely-used synthetic test case in hybrid modeling and simulation-based inference literature [13, 12, 14]. The simulator models the displacement of an ideal, frictionless pendulum, determined by its natural frequency $\omega_0$ and initial angle $\phi_0$. To emulate real observations, we use a damped pendulum model that introduces friction into the system. The damping is controlled by a friction coefficient $\alpha \in \mathbb{R}^+$. The parameters we aim to infer are $\boldsymbol{\theta} = \{\omega_0 \in \left[\frac{\pi}{10}, \pi\right], \ \phi_0 \in [-\pi, \pi]\}$.

**Causal Chambers [34].** Two real, controlled datasets collected from experimental rigs—a wind tunnel and a light tunnel—with adjustable parameters. In the wind tunnel, the target parameter is the hatch opening angle $\boldsymbol{\theta} = \{H \in [0, 45°]\}$. We adopt model A2C3 from [34] as the simulator, which captures pressure dynamics and hatch mechanics, while simplifying aerodynamics and omitting sensor noise, actuator delays, and environmental effects. In the light tunnel, the parameters are the RGB light intensities and a polarizer attenuation factor: $\boldsymbol{\theta} = \{R, G, B \in [0, 255], \ \alpha \in [0, 1]\}$. We use model F3, which simulates photodiode and camera responses under varying exposure and gain, but ignores optical aberrations and sensor noise.

**Real Hemodynamics Data.** This benchmark uses a subset [6, 35] of the MIMIC-II dataset [36], comprising 350 patients who underwent thermodilution—a procedure estimating cardiac output (CO) via cold fluid injection and downstream temperature measurement. Each patient has arterial blood pressure (ABP) signals aligned with CO readings, yielding $\sim 2200$ valid ABP segments with corresponding CO values. For simulation, we use OpenBF [37], a validated 1D cardiovascular flow simulator supporting fast, multiscale finite-volume simulations. The estimated parameters are $\boldsymbol{\theta} = \{\text{HR}, \text{CO}\}$, where HR is heart rate, obtained from ECG measurements. The empirical means and standard deviations of HR and CO in the dataset are $(87, 12)$ beats/min and $(5.1, 1.6)$ L/min respectively. In contrast, the CO from the simulations is derived by the known connection $\text{CO} = \text{HR} \times \text{SV}$, where SV is the stroke volume. HR and SV are sampled from uniform distributions $\text{U}(50, 150)$ beats/min and $\text{U}(40, 140)$ L/beat respectively. This yields CO in the range of $[2, 20]$ L/min. This use case is inherently affected by label noise, since the pairing of HR and CO values with observed pulse waves depends on temporal alignment through patients' electronic records, which is prone to misalignments.

## 4.2 Experimental design

**Baselines.** The primary method we compare against is RoPE [22]. However, a direct comparison on the same test set is unfair, as RoPE is purely transductive. Nevertheless, we include this setting as a baseline, denoted **RoPE full test**. For a fairer inductive comparison, we introduce a single-sample variant of RoPE: for each test point $\boldsymbol{x}_r^{\text{test}}$, we add it to the unpaired training set $\mathcal{D}_u$, compute the OT coupling with simulations $\mathcal{D}_{OT}$, and estimate the posterior using the resulting plan. This is repeated per test point, and we denote this baseline as **RoPE single sample**. We also compare against baselines from [22], including **NPE** (see Section 2.1), applied directly to real observations without domain transfer. To assess the role of the calibration set, we also include an unsupervised domain adaptation (**UDA**) NPE baseline following [38]. In addition, we include OT-only baselines that do not adapt the embedding space. Here, the fixed encoder $h_{\omega^\star}$ is applied to both real and simulated samples, and OT is computed in this space. The full-test variant is denoted **OT-only (full test)**, and the single-sample variant as **OT-only (single sample)**. For all experiments involving OT, we use solvers from the POT Python library [39, 40].

Another baseline is **finetune-only**, where the finetuned encoder $g_{\phi^\star}$ is applied to test samples, and NPE is used to estimate the posterior directly from $g_{\phi^\star}(\boldsymbol{x}_r)$, without OT-based mixing.

Finally, we include two additional baselines: the upper bound **SBI**, where NPE is trained and tested on simulations, and the **prior** estimator.

**Metrics.**    We evaluate our method using the same metrics as in [22], which assess two critical aspects of posterior estimation: accuracy and calibration.

The first metric is the **log-posterior probability (LPP)**, defined as the average log-likelihood of the true parameter values under the estimated posterior distribution. It measures how much density the true parameters receive in their estimated posteriors, effectively capturing the sharpness and accuracy of the model's predictions. Higher LPP indicates a better match between the estimated posterior and true value.

The second metric is the **average coverage area under the curve (ACAUC)**, which reflects how well the model's credible intervals align with the true parameter coverage. It provides a measure of calibration by comparing the fraction of true parameters falling within the estimated credible intervals at different confidence levels. In all experiments, credible intervals are obtained by drawing 1000 samples from the corresponding approximate posterior distribution. A perfectly calibrated model has an ACAUC of zero, while positive values indicate overconfident estimates and negative values indicate underconfident estimates. For a detailed mathematical definition, we refer the reader to [22].

## 4.3   Results

In this section, we present the key findings of our experiments.  A detailed description of the implementation can be found in the supplementary material.

### 4.3.1   Performance Across Different Calibration Set Sizes

We evaluate methods that rely on calibration data using 5-fold cross-validation.  In each fold, a different, randomly sampled subset of the calibration data is used for training and validation, with independent random initialization of model weights. This approach captures both data variability and the effects of random initialization, providing a robust assessment of performance.

We evaluate calibration set sizes of 10, 50, 200, and 1000 samples, while keeping the test set size fixed at 1000 samples across all benchmarks. The simulation set, $\mathcal{D}_{SBI}$, used to train both the NSE and NPE, as well as the unpaired real observations set, $\mathcal{D}_u$, each contain 1000 samples. Similarly, the set of simulations used for the OT in RoPE and our proposed joint training, $\mathcal{D}_{OT}$, also consists of 1000 samples. For training the amortised posterior estimator (Section 3.2), we exclusively use $\mathcal{D}_u$. Following the guidelines in [22], the entropy regularization weight $\gamma$ is set to 0.5 for all baselines involving OT, including our joint training approach. Finally, since this section focuses on the setting where both simulations and real observations are drawn from the same prior distribution $p(\boldsymbol{\theta})$, we use balanced Sinkhorn OT for RoPE, effectively setting $\rho$ in eq. 1 to a very high value.

The results, including mean scores and standard deviations computed across folds, are presented in Fig. 2. Our method (solid red line) consistently outperforms the transductive single-sample RoPE baseline (solid orange line) in terms of LPP, while maintaining ACAUC close to zero across most calibration set sizes. This indicates that it achieves a robust **inductive** parameter estimation compared to RoPE. With larger calibration sets, our method matches or even exceeds the performance of the full-test RoPE (solid green line), which has access to the entire test set—highlighting the effectiveness of combining OT-based domain transfer with supervised calibration. It also highlights the importance of incorporating a calibration set, even a small one, as the UDA baseline completely fails under significant misspecification, consistent with the observations in [41]. However, as the calibration set size increases, our method—like the fine-tuning baselines—shows a decline in confidence (lower ACAUC). In contrast, the RoPE variants are less affected by this trend, suggesting that our joint training approach increasingly relies on the supervised loss over the OT loss as more calibration data becomes available. We also evaluate performance on two additional benchmarks exhibiting more moderate misspecification (Appendix C.1), also considered in [22]. Similar to RoPE, FRISBI shows no significant advantage over baseline methods in simpler and minimally misspecified setting.

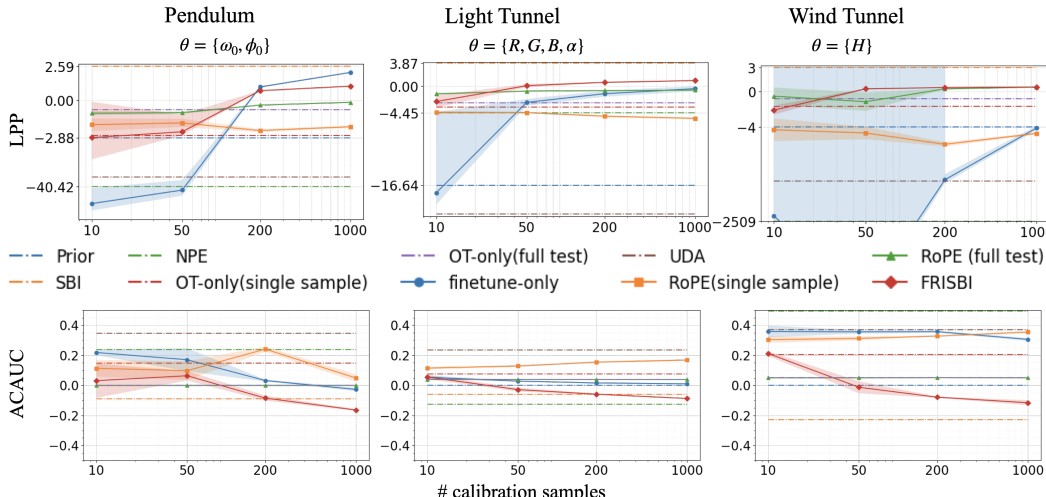

Figure 2: **Results across different calibration set sizes.**. The **top** row displays performance in terms of LPP ($\uparrow$) while the **bottom** one is the calibration metric ACAUC ($\to 0 \leftarrow$). The horizontal axis indicates the sample size while the vertical one is the metric value. Baselines that do not rely on a calibration set are represented by fixed horizontal dashed lines for easier comparison.

### 4.3.2 Ablation

**The importance of joint training.** As described in Section 3, our full pipeline consists of two main components: joint distribution and point-to-point alignment, which we refer to as **joint training only**, and the posterior-mixture amortization step, which we refer to as **amortized solution only**. To assess the individual contributions and necessity of these components, we evaluate their performance separately and in combination. For the **amortized solution only** baseline, we train the amortization step described in Section 3.2 directly to approximate the posteriors estimated by RoPE, using the unpaired real set $\mathcal{D}_u$ and the OT simulations set $\mathcal{D}_{OT}$. For the **joint training only** baseline, we evaluate the posteriors obtained through our joint training approach described in Section 3.1, without the additional amortization step. Finally, we compare these to the **full pipeline**, which combines both components as described in Section 3.

The mean scores and standard deviations for each variant are reported in Fig. 3. Overall, the joint training approach and the full pipeline achieve comparable performance in terms of LPP and ACAUC, both consistently outperforming the amortized version of the transductive RoPE solution (blue). This suggests that the performance gains in our pipeline are largely attributable to the joint training strategy. The full pipeline offers the added benefit of not requiring access to the simulations $\mathcal{D}_{OT}$ at test time. Notable differences arise in specific cases: the full pipeline (green) performs better in the low-calibration regime of the light tunnel benchmark (middle of Fig. 3), suggesting it may act as a regularizer for the posterior mixture when calibration data is limited, while joint training alone (orange) outperforms the full pipeline in the pendulum benchmark. In the latter, we observed that in one fold, the cNF model used to amortize the posterior mixture failed to reduce its training loss (Eq. 5), indicating difficulties in learning the posterior. This could potentially be mitigated by using a more expressive cNF model.

**Hyperparameter sensitivity analysis.** In Appendix C.2, we present a sensitivity analysis of the hyperparameters $\gamma$ and $\lambda$ on the Light Tunnel benchmark. As in RoPE, a larger $\gamma$ is beneficial when the calibration set is small, producing a more diffuse OT coupling. With larger calibration sets (e.g., 1,000 samples), $\gamma$ can be reduced to obtain a sharper coupling. The effect of $\lambda$ is evaluated under 10% label noise, since noise influences the weighting of the supervised objective. Higher $\lambda$ values decrease confidence as the calibration set grows, while smaller values yield better LPP scores, suggesting that an adaptive tuning of $\lambda$ based on noise level and data size may be advantageous.

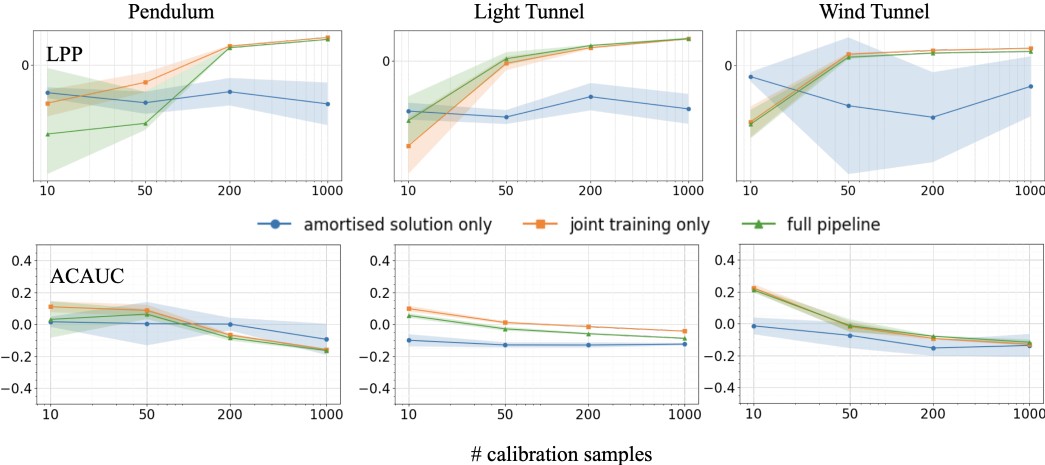

Figure 3: **Ablation Analysis.** Comparison of joint training only (3.1), solution amortization only (3.2), and the full pipeline. The horizontal axis shows the number of calibration samples, while the vertical axis represents the LPP($\uparrow$, top) and ACAUC($\to 0 \leftarrow$, bottom) scores.

### 4.3.3 Label Noise Robustness

In a more realistic scenario, the calibration set known parameters $\theta$ are themselves measurements and thus inherently noisy. For example, in the case of CO (cardiac output) labels, both the measurement procedure and the temporal alignment with the corresponding ABP (arterial blood pressure) signal can introduce noise. This means that the resulting simulated observations $x_s$ may also be inaccurately paired with the real observations, potentially affecting the quality of the calibration set.

To evaluate the robustness of our proposed method to label noise, we add Gaussian noise to the calibration labels, corresponding to 1% and 10% of the parameter range of the assumed prior in the light tunnel benchmark. The models are trained using these noisy labels, while the test set remains clean for evaluation, as in the previous subsections. In this experiments we still consider the balanced OT settings (no prior mismatch) and keep $\gamma$ at 0.5. In Fig. 4, our method (solid red line) almost consistently achieves higher LPP scores than the single-sample RoPE baseline (solid orange line), indicating greater robustness to label noise due to its inductive nature. While it is somewhat more sensitive to high noise levels compared to the full-test RoPE baseline (solid green line), this sensitivity diminishes with larger calibration sets. For example, with 200 calibration samples, the performance gap noticeably narrows.

In the CO estimation experiment, we explicitly account for the increased uncertainty in the calibration set by setting the entropic regularization parameter $\gamma$ to a higher value of 1.5 for the RoPE and OT baselines and in our proposed method. Additionally, in the RoPE baselines we use unbalanced OT settings to handle known prior mismatches, as suggested in [22]. The calibration set size in these experiments is 200, and we report results across 5 random splits to assess robustness. It is important to note that clean test-set $\theta$ labels are not available in this setting. As shown in Fig. 5, Our method (in red) significantly outperforms all baselines, including RoPE, in terms of LPP, while showing slight overconfidence (ACAUC < 0.2), highlighting the inference benefits of our amortized inductive framework in real complex settings.

## 5 Discussion and Conclusion

In this work, we introduce an amortized and inductive posterior estimator for misspecified simulation-based inference. Our method leverages mini-batch optimal transport to enable joint training over both unpaired and small paired calibration sets. The final posterior, approximated by an OT-based mixture, is amortised by a conditional normalizing flow, eliminating the need for additional transport computations or access to simulations at test time – a key limitation of transductive approaches like RoPE. Our approach demonstrates competitive performance across a

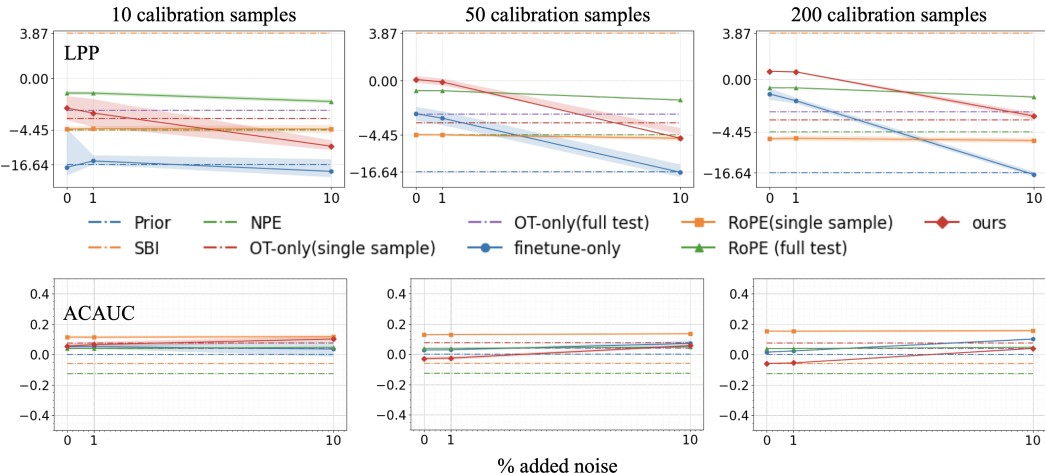

Figure 4: **Label Noise Robustness.** Impact of increasing label noise on performance, measured by LPP ($\uparrow$, top) and ACAUC ($\to 0 \leftarrow$, bottom), across three calibration set sizes (noted above each panel) on the light tunnel benchmark. The horizontal axis represents the noise rate, while the vertical axis shows the metric score.

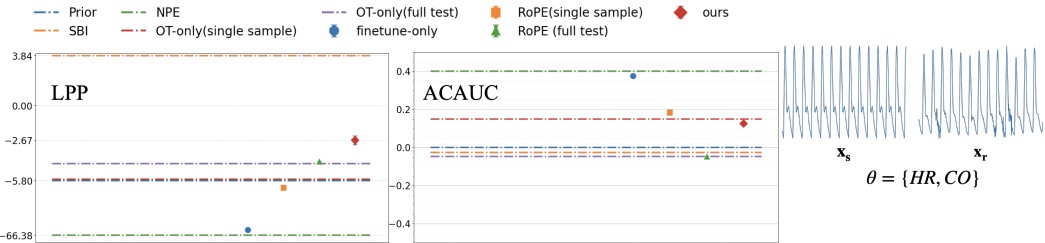

Figure 5: **Cardiac Biomarker Estimation.** Performance comparison across all baselines for heart rate (HR) and cardiac output (CO) estimation, using a calibration set of 200 samples. On the right, an example of real and simulated arterial pulse waveforms is shown.

range of benchmarks, including both synthetic and complex real-world datasets.

**Dimensionality and scalability.** While our experiments focus on relatively low-dimensional cases representative of many real-world scenarios, our framework naturally extends to higher-dimensional settings. Two main challenges arise: (i) the mismatch between simulated and real embeddings tends to increase with dimensionality, requiring larger calibration sets, and (ii) the embedding dimension needed to capture sufficient statistics typically grows with the parameter dimension, complicating the coupling. Following Chen et al. [42], we use an overparameterized embedding dimension of 16 to mitigate this issue. The joint optimization of supervised and OT objectives further alleviates the limitations of small calibration sets by exploiting unlabeled data. As discussed in Section 3.1, FRISBI is expected to scale more efficiently than RoPE in high-dimensional settings.

**Limitations and future directions.** Sensitivity to label noise, as observed in our experiments, suggests that training data quality significantly impacts posterior accuracy. While this could be partially mitigated with a higher entropy regularization weight, more adaptive joint loss formulations should be explored and specifically adaptive updates of the supervised loss weight $\lambda$. Additionally, active learning or uncertainty-aware sampling strategies could be investigated to guide calibration set selection under a fixed budget. Finally, the training process of our proposed method still involves two separate stages: the joint OT-supervised training and the subsequent amortization of the induced posterior mixture. A more holistic alternative could involve learning the transport mapping directly through a neural OT framework, potentially integrating the matching and inference stages into a single unified process.

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

## A  Datasets clarification

Table 1: Summary of datasets used in the pipeline.

| Dataset | Observations | $\theta$ Labels | Usage |
|---|---|---|---|
| $D_{\text{SBI}}$ | simulated ($\boldsymbol{x}_s$) | yes | train NPE, NSE |
| $D_{\text{OT}}$ | simulated ($\boldsymbol{x}_s$) | no | train over the OT objective in 3.1, amortization of posterior mixture in 3.2 |
| $D_{\text{calib}}$ | real ($\boldsymbol{x}_r$) | yes | train 3.1, MSE objective (and OT) |
| $D_u$ | real ($\boldsymbol{x}_r$) | no | train 3.1, OT objective, amortization of posterior mixture in 3.2 |

## B  Semi-balanced Optimal transport solvers.

We develop next how to solve for semi-balanced entropic optimal transport problems discussed in Sections 2.2 and 3. The overall problem reads as

$$\boldsymbol{P}^\star = \arg\min_{\boldsymbol{P}\in\mathcal{B}(N_{test},N_{OT})} \mathcal{L}(\boldsymbol{P}) := \langle \boldsymbol{P},\,\boldsymbol{C}\rangle + \rho\,\mathrm{KL}\big(\boldsymbol{P}^\top \mathbf{1}_{N_{test}} \,\|\, \tfrac{1}{N_{OT}}\mathbf{1}_{N_{OT}}\big) + \gamma\,\langle \boldsymbol{P},\log\boldsymbol{P}\rangle. \quad (6)$$

This type of problem can be generally solved using an iterative bregman projection solver [31, 32, 29, 33], or equivalently mirror-descent algorithms following the KL geometry. It comes down to the following steps:

Step 1. Compute the gradient of the objective function

$$\nabla\mathcal{L}(\boldsymbol{P}^{(t)}) = \boldsymbol{C} + \rho\mathbf{1}\log(N_{OT}\boldsymbol{P}^{(t)\top}\mathbf{1})^\top + \gamma\log\boldsymbol{P}^{(t)} \quad (7)$$

step 2. Then for a given learning rate $\tau$, one has to solve the problem

$$\boldsymbol{P}^{(t+1)} \leftarrow \arg\min_{\boldsymbol{P}\in\mathcal{B}(N_{test},N_{OT})} \langle\nabla\mathcal{L}(\boldsymbol{P}^{(t)}),\boldsymbol{P}\rangle + \tau KL(\boldsymbol{P}|\boldsymbol{P}^{(t)}) \quad (8)$$

we have

$$
\begin{aligned}
&\langle\nabla\mathcal{L}(\boldsymbol{P}^{(t)}),\boldsymbol{P}\rangle + \tau KL(\boldsymbol{P}|\boldsymbol{P}^{(t)})\\
={}&\langle\boldsymbol{C}+\rho\mathbf{1}\log(N_{OT}\boldsymbol{P}^{(t)\top}\mathbf{1})^\top + \gamma\log\boldsymbol{P}^{(t)},\boldsymbol{P}\rangle + \tau\langle\boldsymbol{P},\log\boldsymbol{P}-\log\boldsymbol{P}^{(t)}\rangle\\
(\text{setting }\gamma=\tau)={}&\langle\boldsymbol{C}+\rho\mathbf{1}\log(N_{OT}\boldsymbol{P}^{(t)\top}\mathbf{1})^\top,\boldsymbol{P}\rangle + \gamma\langle\boldsymbol{P},\log\boldsymbol{P}\rangle \quad (9)\\
={}&\langle-\log e^{\frac{-\boldsymbol{C}-\rho\mathbf{1}\log(N_{OT}\boldsymbol{P}^{(t)\top}\mathbf{1})^\top}{\gamma}},\boldsymbol{P}\rangle + \gamma\langle\boldsymbol{P},\log\boldsymbol{P}\rangle\\
={}&\gamma KL(\boldsymbol{P}|\boldsymbol{K}_\rho^{(t)})
\end{aligned}
$$

with $\boldsymbol{K}_\rho^{(t)} = e^{\frac{-\boldsymbol{C}-\rho\mathbf{1}\log(N_{OT}\boldsymbol{P}^{(t)\top}\mathbf{1})^\top}{\gamma}}$. Hence this problem comes down to a KL projection on the set $\mathcal{B}(N_{test},N_{OT})$ of the Gibbs kernel $\boldsymbol{K}_\rho^{(t)}$. As detailed in Proposition 1 in [31]) this problem admits a close-form solution detailed in Equation 2.

## C  Additional experiments

### C.1  CS and SIR benchmarks

We follow the evaluation protocol of [22], originally introduced by [43].

**CS.** The cancer-stromal cell simulator models 2D cell growth with three Poisson rate parameters $(\lambda_c,\lambda_p,\lambda_d)$. Each sample includes cell counts and the mean and maximum distance between stromal and nearest cancer cells. Misspecification is induced by removing cancer cells located too close to their parent.

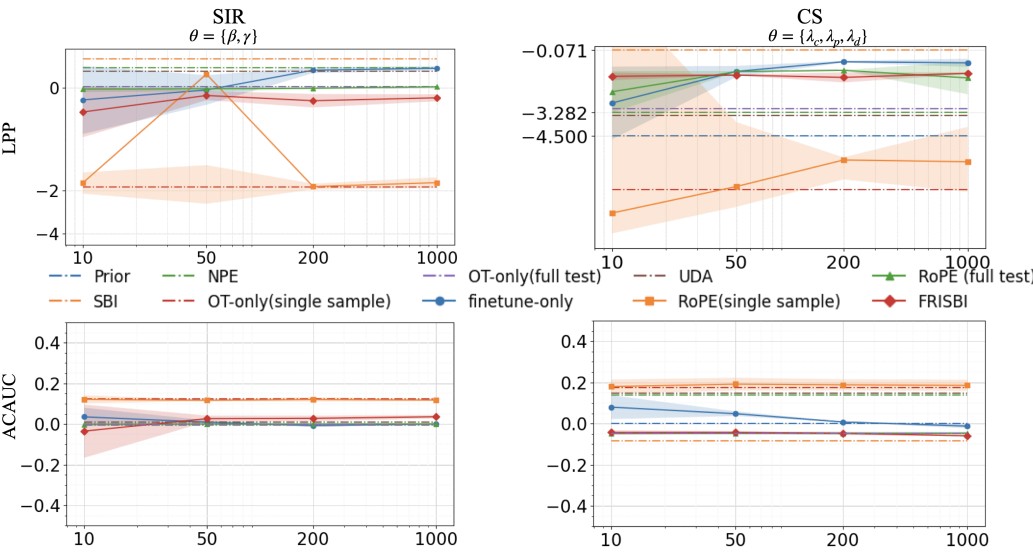

Figure 6: **Minimally misspecified benchmarks (CS and SIR)**. The horizontal axis shows the number of calibration samples, while the vertical axis represents the LPP($\uparrow$, top) and ACAUC($\rightarrow 0 \leftarrow$, bottom) scores.

**SIR.** The stochastic epidemic model simulates infection and recovery dynamics with rates $(\beta, \gamma)$. Observations comprise summary statistics of infection counts, timing, and autocorrelation. Misspecification is introduced by delaying weekend infections, adding $5\%$ of them to the following Monday.

We observe that both FRISBI (red) and RoPE underperform relative to the vanilla NPE baseline (dashed green) in the SIR benchmark and achieve performance comparable to the finetuning-only baseline (blue) in the CS case. This suggests that, under minimal misspecification, a standard NPE is sufficient, and in simpler systems, even a small calibration set can adequately capture the mapping between observations and parameters.

## C.2 Sensitivity analysis w.r.t hyperparameters $\gamma$ and $\lambda$

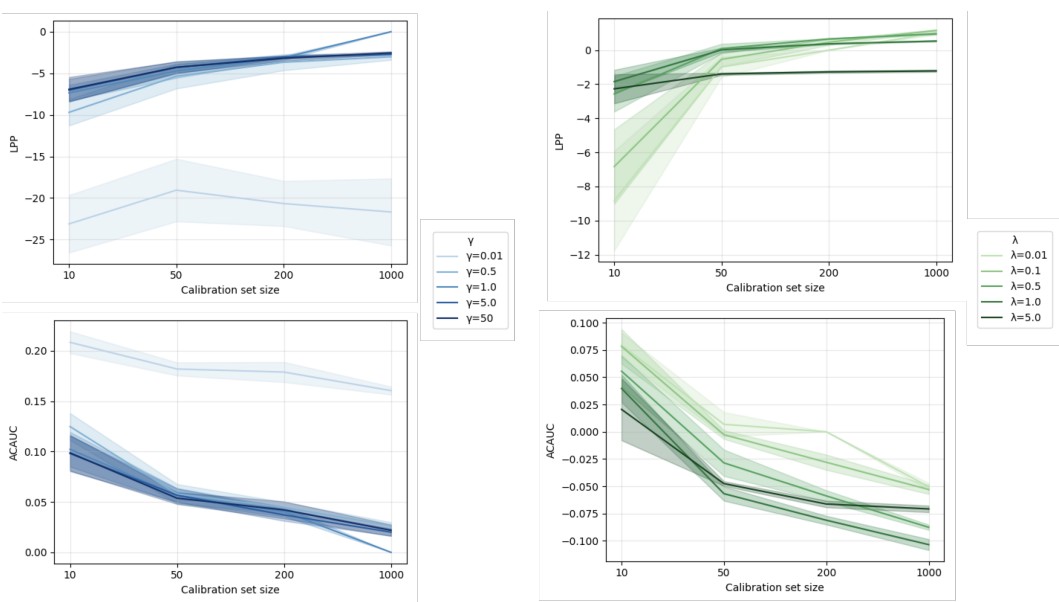

Figure 7: **Sensitivity analysis of hyperparameters** $\gamma$ **and** $\lambda$**.** Both experiments are conducted on the Light Tunnel benchmark. The $\lambda$ sensitivity analysis (right) is performed under a 10% label noise setting. The horizontal axis shows the number of calibration samples, while the vertical axis reports the LPP ($\uparrow$, top) and ACAUC ($\rightarrow 0 \leftarrow$, bottom) scores. The legend indicates the different $\gamma$ and $\lambda$ values considered and their corresponding shades.

