# OpenReview forum: "Inductive Domain Transfer In Misspecified Simulation-Based Inference"
_NeurIPS.cc/2025/Conference — NeurIPS 2025 poster_

### Official Review · Reviewer_jspK · 2025-06-11

**Clarity:** 2
**Significance:** 2
**Originality:** 2
**Rating:** 4
**Confidence:** 4

**Summary:**

This paper introduces a method for addressing misspecification in SBI. The method builds on RoPE, a recently published method that addresses the same challenge, and eliminates RoPE's need for a calibration set of real observations during inference. This is achieved by 1.) incorporating the callibration set into the fine-tuning objective of the pretrained NPE embedding network, and 2.) by adapting the optimal transport objective to allow for inductive inference requiring only a single observation. Finally, the authors propose to train an additional normalizing flow to amortize the computation of the inference-time OT plan.
The authors evaluate their method in comparison to RoPE and various ablations mainly using the tasks, metrics, and experiments considered in the RoPE paper.

**Questions:**

- Can the authors provide some mathematical justification why the $\rho=0$ transport plan is a good choice, and what posterior one obtains in the end? Does the self-calibration property of RoPE or something similar still hold when $\rho=0$?
- How should users tune $\lambda$?
- Have the authors considered different types of regularization other than the OT term in Eq. (4)? Replacing or dropping it would eliminate the dependence on the callibration set of observations during finetuning. How exactly does the OT term compensate, if the supervised term is off?
- Since their paper is so closely related to RoPE, could the authors also apply their method to the CS and SIR task? Or was there a good reason not to?
- Some numbers that should be consistent between this paper and the RoPE paper seem different? Why is that? E.g. the LPP for the SBI upper bound on the light tunnel task.

**Ethical Concerns:**

["NO or VERY MINOR ethics concerns only"]

**Final Justification:**

The authors satisfactorily addressed many of my concerns, so I am raising my score from 3 to 4. Overall, this paper brings modest improvements over RoPE, both performance-wise and from a conceptual point of view (transductive vs. inductive). However, similar to RoPE, the method's dependence on a calibration set and its overall complexity, requiring many training and tuning steps, are key limitations that restrict applicability and impact.

**Limitations:**

Yes.

**Paper Formatting Concerns:**

No major formatting issues.

**Quality:**

2

**Strengths And Weaknesses:**

This paper tackles the important challenge of handling simulator misspecification in SBI.

## Strengths
- The method addresses some proper limitations of RoPE, particularly its dependence on a calibration set of real observations during inference.
- The authors consider many ablations and include an experiment concerning label noise robustness. This is highly relevant in the RoPE setting, where access to pairs of real observations and ground truth parameter measurements (which might be noisy) is assumed.

## Weaknesses

### OT with $\rho = 0$:
To make their method inductive rather than transductive, the authors drop the column-marginal constraint of the semi-balanced OT objective used in RoPE. This enables them to obtain a closed-form transport plan that does not depend on other observations. In particular, they write:
> While being more computationally intensive for training, such approaches are also known for leading to biased OT solutions and a significant sensitivity to the chosen batch size, hence further motivate our choice for $\rho = 0$.

However, the claim about biased OT solutions and the general OT framing is slightly baffling, as completely dropping the column-marginal constraint (which is $\rho=\infty$ for balanced OT) results in something that basically ceases to be optimal transport?
Rather, the "transport plan" the authors obtain with $\rho = 0$ is a normalized squared exponential kernel that weights certain simulations higher than others based on the distance between their embedding and the observation's embedding.

This might be a reasonable thing to do, but the authors should more clearly describe that this is what they do, and explain why this might be a reasonable thing to do.

### Finetuning objective:
The $\rho =0$ transport plan is also used in the finetuning objective, combined with the supervised loss used in RoPE. However, it's unclear what the exact role of the OT part in the objective is. The authors write:
> In contrast, when calibration pairs are noisy or mismatched, the two objectives may conflict, allowing the term to compensate for uncertainties in the calibration set.

The paper does not discuss how this compensation works, especially given the special transport plan.

### Complexity and hyperparameter tuning:
While the authors mention this limitation, the method includes quite a few complex steps, which hinders applicability.
Users need to first train an NPE flow and an embedding net. Then, the finetuning of the embedding net is based on a modified objective, which introduces another hyperparameter $\lambda$ (beyond the hyperparameter for the entropy regularization $\gamma$).
Finetuning is followed by training another flow to amortize over the computation of the transport plan.

The discussion of how to choose these hyperparameters, particularly $\lambda$, and their effect on inference is insufficient.
Furthermore the method still assumes access to two different calibration sets.

### Experiments:
While the authors consider many different ablations, the paper does not consider all the tasks performed in RoPE.

---

> ### Author Rebuttal · Authors · 2025-07-31
>
> Thank you for the constructive and detailed review, and for raising important points to clarify in our work. Below, we address them point by point.
>
> > W1+Q1. **OT with $\rho=0$**
>
> Thank you for this interesting analysis. First, we would like to clarify that the inductiveness of our approach is not inherently due to our choice of $\rho=0$, but rather from how we optimize $g_{\phi}$ (Stage 1) and $q_{\xi}$ (Stage 2), allowing inference without solving an OT problem at test time. Our learning scheme can naturally be extended to $\rho > 0$ by computing an optimal $P_{ij}$ in Stage 1 (Line 7) using RoPE’s OT solver, and integrating the computation of $\alpha_{ij}$, and thus the OT plan, by batch for the training of Stage 2. This process remains inductive. We will clarify this in L129–140.
>
> Second, we believe that one commonly used term for the matching problem in Equation 1 is the one used in L88: "semi-balanced entropic OT problem" (though “semi-relaxed” or “semi-unbalanced” are also used), following recent works like [30], [A, B] with $\rho = 0$ and [C] with $\rho > 0$. This corresponds to a specific instance of unbalanced OT, with $\infty$ as hyperparameter on the first marginal and $\rho$ on the second. After the introduction, we simplified terminology to "OT" or "OT coupling" for conciseness. If the reviewer finds it important to be more specific throughout the text, we welcome precise suggestions.
>
> Finally, we agree with the reviewer’s point on behavior with $\rho = 0$. [D] offers good intuitions on the semi-balanced OT loss without entropic regularization. They show that minimizing the support and masses of the target distribution for a fixed input one (i.e., finding a minimal Wasserstein estimator) generalizes k-means depending on the inner cost. Entropic regularization on $\mathbf{P}$ naturally yields a soft k-means. These problems still differ from ours, as the target support is $\{ h_{w^\star}(x_s^j) \}$, and we optimize $g_{\phi}$ over inputs $\{x_r^i\}$. Therefore, embeddings from simulations act as prototypes (or clusters) over which we fit real sample embeddings. Setting $\rho > 0$ would encourage uniform use of these prototypes for fitting real data (whose validity is questionable in mini-batch settings). However, as mentioned, $\rho = 0$ scales better than $\rho > 0$ thanks to the closed-form; the OT solution does not depend on the sampled batch and provides empirically strong performance. We will clarify this analysis and the referenced arguments in the revised manuscript.
>
> [A] Clark, R. A., Needham, T., & Weighill, T. (2025, April). Generalized dimension reduction using semi-relaxed Gromov-Wasserstein distance. In Proceedings of the AAAI Conference on Artificial Intelligence (Vol. 39, No. 15, pp. 16082–16090).
>
> [B] Van Assel & al. (2025). Distributional Reduction: Unifying Dimensionality Reduction and Clustering with Gromov-Wasserstein. Transactions on Machine Learning Research.
>
> [C] Van Assel, H., & Balestriero, R. (2024). A graph matching approach to balanced data sub-sampling for self-supervised learning. In NeurIPS 2024 Workshop: Self-Supervised Learning—Theory and Practice.
>
> [D] Canas, G., & Rosasco, L. (2012). Learning probability measures with respect to optimal transport metrics. Advances in Neural Information Processing Systems, 25.
>
>
> > W2. **Finetuning objective**: clarifying joint OT+supervised objective
>
> The idea was that, unlike in RoPE where the finetuning is done first, then $g$ is fixed and the OT coupling is applied to the embeddings obtained by $g$, in our framework we propose to train $g$ jointly on both the supervised objective (using the calibration set) and the OT objective (using both the calibration set and a large unsupervised set of real observations). The motivation is that, since the calibration set is small and we want to avoid overfitting by the encoder $g$, the OT objective act as a regularizer—especially given the large number of unpaired real observations it is trained on. We will make the quoted sentence clearer in the revised version.
>
>
>
> > W3. + Q2. **Complexity and hyperparameter tuning:**
>
> - We would like to clarify that the NPE training stage, followed by the finetuning of $g$, is shared between RoPE and our method. The only additional step we introduce is the amortization stage, which is specifically designed to remove the need for simulation access at test time, a requirement that remains in RoPE, even in its "inductive" variant (RoPE-single sample).
>
> - Hyperparameter selection: In this regard, our approach does not differ significantly from RoPE. In their work, the authors suggest that practitioners, if possible, hold out a validation set of labeled real observations from the calibration set to tune $\gamma$ (and $\rho$), or otherwise rely on default values informed by a sensitivity analysis. Similarly, we provide a sensitivity analysis with respect to $\gamma$ (please refer to W3 in the reply to reviewer TtjD ) and $\lambda$ on the light tunnel benchmark (w.r.t LPP).
>
>
> | num_samples in calib. set ↓ / λ → | 0.01            | 0.5             | 1.0             | 5.0             | 50              |
> |---------------------|------------------|------------------|------------------|------------------|------------------|
> | 10                  | -23.12 ± 3.45    | -9.70 ± 1.59     | -7.35 ± 0.94     | -7.04 ± 1.36     | -6.93 ± 1.48     |
> | 50                  | -19.06 ± 3.76    | -5.36 ± 1.47     | -4.72 ± 0.83     | -4.30 ± 0.73     | -4.28 ± 0.68     |
> | 200                 | -20.67 ± 2.72    | -3.67 ± 0.97     | -3.13 ± 0.27     | -3.06 ± 0.15     | -3.19 ± 0.32     |
>
> The $\lambda$ parameter was analyzed on the light tunnel benchmark with synthetic noise added at a 10% level. We observe that as long as $\lambda > 1$, the results remain relatively stable.
>
> - Calibration set: When we refer to the calibration set, we mean the small set of labeled real observations $(x_r, \theta)$. In both our framework and the RoPE baselines, there is only one calibration set, which is used to train the supervised objective. We kindly ask the reviewer to point us to the source of this confusion so that we can clarify it appropriately in the revised version.
>
> Questions:
>
> > Q1. $\rho=0$ discussion, posterior computation, self-calibration.
> - For the discussion about the choice of $\rho=0$ please refer to W1.
> - The posterior approximation in Stage 1 for $\rho = 0$ remains unchanged and is expressed as a mixture of simulated posteriors, weighted by the optimal transport plan coefficients: $\tilde{p}(\theta \mid x_r^i) := \sum_{j=1}^{N_{\text{OT}}} \alpha_{ij} \, q_{\psi^\star}(\theta \mid h_{\omega^\star}(x_s^j))$, where $\alpha_{ij} := N_u P_{ij}^\star$. Setting $\rho = 0$ does not alter the assumptions underlying RoPE’s equation (Eq. 4):
> $p(\theta \mid \mathbf{x}_r) = \int p(\theta \mid \mathbf{x}_s) \pi^{\star}(\mathbf{x}_s \mid \mathbf{x}_r) d\mathbf{x}_s$, but only changes how the distribution $\pi^\star$ is approximated.
> - Indeed, we cannot guarantee the self-calibration property as in RoPE (balanced RoPE without the relaxation of the column-marginal constraint). Since in our case we don't constrain the column-marginal at all, we cannot guarantee that $\sum_{i=1}^{N_u} P_{ij}^\star = \frac{1}{N_{OT}}$, so the derivation of the self-calibration property of balanced RoPE (RoPE appendix, p. 20) does not hold. However, in practice RoPE also relaxes this constraint, so self-calibration is not guaranteed as well.
>
>
> >Q2. How should users tune $\lambda$ ?
>
> Please refer to W3.
>
> >Q3. OT+supervised terms purpose
>
> The calibration set $D_{calib}$ is a small supervised set of pairs $(\theta, x_r)$, from which we can simulate the corresponding $x_s = S(\theta)$ (with $S$ being the simulator). It is used to train the supervised objective, specifically, to finetune the encoder $g$ so that it embeds $x_r$ as close as possible to its corresponding $x_s$.
> The entropic OT term compensates for the uncertainty in the supervised matching, which arises due to the limited size of the supervised calibration set. It enables a soft matching between $g(x_r)$ and $h(x_s)$, and also allows us to train on a large unlabeled set of real observations $x_r$, denoted as $D_{OT}$. As a result, $g$ is optimized jointly on both the supervised and OT objectives.
> We observe that the pure OT baselines perform worse than both RoPE and our proposed method. Additionally, in the $\lambda$ sensitivity experiment shown above, we demonstrate that heavily down-weighting the supervised loss (e.g., $\lambda = 0.01$) leads to significantly degraded performance.
>
> >Q4. CS and SIR tasks
>
> We chose not to analyze the CS or SIR tasks, as the RoPE paper indicated that the level of misspecification in these cases was limited and vanilla NPE or OT without a calibration set already performed well. Instead, we introduced a more complex use case: the real cardiac biomarker inference task from the arterial pressure wave benchmark. This represents a substantially different challenge compared to the RoPE cardiac biomarkers task, which was entirely synthetic.
>
> >Q5.  results consistency
>
> We did not have access to the RoPE code, so we implemented our own version. Naturally, this may lead to some variations in results. However, the main reason for the differences in baseline scores is the use of batch normalization.
> As noted in the implementation details (p. 22–25, appendix) of the RoPE paper, they do not use batch normalization in the NSE encoders, which results in a very low LPP for the vanilla NPE baseline in the highly misspecified benchmarks. To ensure a fair comparison, we added batch normalization to all baselines in our implementation (we will mention it in the revised manuscript).

---

> ### Comment · Reviewer_jspK · 2025-08-02
>
> Thank you for your clarifications regarding the inductiveness of the approach and hyperparameter selection. Apologies for misusing the term "calibration set", I am aware that there is only one. The other set I was referring to was $D_{unpaired}$. The paper is sufficiently clear on this point.
>
> Thanks for the references regarding the semi-balanced OT with $\rho=0$. This addresses my concerns regarding terminology. Whether $\rho=0$ is considered to be semi-balanced OT seemingly depends on the setting (another example is this paper, where again $\rho>0$ [1]).
> I still think the method the authors end up with can be framed without any OT (kernel weighted posterior mixture), and the paper would benefit from at least discussing this, in my view, *simpler* framing.
>
> [1] Janati, H., Muzellec, B., Peyré, G., & Cuturi, M. (2020). Entropic optimal transport between unbalanced gaussian measures has a closed form. Advances in neural information processing systems, 33, 10468-10479.
>
>
> Regarding the experiments:
>
> First, I'm not convinced by the author's reasoning to not include SIR and CS (may it be in the appendix). For a paper that is so closely aligned with or based on another paper (RoPE), testing on the full benchmark suite proposed by said paper would be valuable.
>
> Specifically, the argument that these tasks are not "misspecified enough" is questionable. Wouldn't it be good to see that the proposed method still works well even if the level of misspecification is not large? Do users always know the level of misspecification beforehand? A method tackling misspecification should produce reasonable results even if the simulator is not, or only slightly, misspecified, no?
>
> Second, it is unfortunate that RoPE does indeed not seem to provide code, and this is certainly not the author's fault. I agree that some variations in results may be expected when reimplementing a method.
>
> However, for some settings/experiments the differences are quite large? And why would the use of *batch normalization* in encoders have such a large effect on the LPP? I really don't understand this. Could the authors explain? Are there other differences in implementation to the ones listed in the RoPE appendix?

---

> > ### Author Response · Authors · 2025-08-05
> > **1/2**
> >
> > > W1-Q1 discussion : Method framing
> >
> > We agree with the reviewer that a discussion highlighting our simplified framework w.r to the full OT formulation in RoPE should be added to the paper. However, we believe it is important to retain the OT framing, as it plays a crucial role in clarifying the relationship between our work and RoPE. As agreed in our previous response, we plan to include this discussion in the revised manuscript. Thank you for pointing us to the referenced paper,  it is indeed relevant and will be included in the expanded discussion.
> >
> > > Less misspecified benchmarks (CS and SIR)
> >
> > The reviewer raises a valid point, and we agree that it is important to include these benchmarks as well. As a preliminary sanity check, we conducted experiments with the SIR benchmarks. Please note that some variations in the data generation process could influence the results. We observe that, indeed, the vanilla NPE and finetuning-only baselines outperform all OT-based baselines, including ours. Our method shows slightly lower LPP than full RoPE, but still performs better than the single-sample RoPE variant.
> >
> > Additionally, the size of the calibration set does not appear to have a significant impact on the LPP score, a trend also reported in RoPE (Fig. 1). In this setting, the OT-based posterior estimation seems to be an over-correction, leading to a less accurate posterior. This is particularly true for entropic OT, which produces more diffused couplings as the entropy increases, suggesting that lower values could lead to improvement as well as an \underline{exact} OT counterpart.
> >
> > In practice, the degree of misspecification is unknown, but it may be possible to estimate it using the calibration set. This represents a promising direction for future research and could be linked to the previously discussed selection of the hyperparameters $\lambda$ and $\gamma$. We will include this discussion in the revised manuscript.
> >
> > LPP scores calculated for the SIR benchmark ($\lambda=1, \gamma=0.5$): **posted in a second reply due to characters constraints**
> >
> >
> > **ACAUC:** All baselines exhibit values close to 0, except for the single-sample variants, which show overconfidence. Our method is slightly overconfident (0<$\text{ACAUC} < 0.1$), while RoPE and the OT-only baseline are slightly underconfident ($-0.1 < \text{ACAUC}<0$).
> >
> >  > different baseline results from RoPE
> >
> > While batch normalization is known to improve training dynamics, it has also been shown to enhance out-of-distribution generalization to some extent [1]. In fact, in several implementations of SBI, the encoder for the summary statistics includes batch normalization (e.g., in astrophysics [2,3]), a practice we also follow in our work. As a sanity check, we ran the SBI and NPE baselines without batch normalization and obtained LPP of -2.23 and -59 (vs. 3.87 and -4.45 with BN, reported in our paper), respectively, on the Light Tunnel benchmark. This confirms that removing batch normalization degrades LPP performance for these baselines.
> >
> > Additionally, there may be other differences between our implementation and RoPE’s, such as the architecture of the density estimator. We used the MAF implementation from the `nflows` Python library [4], while RoPE employed UMNN-MAF. Other variations may exist, for example, in the choice of OT libraries (we used `POT`, while RoPE used `OTT`). However, we emphasize that, similar to RoPE, our contribution is a general framework rather than a specific architecture or algorithm. To fairly demonstrate its effectiveness, we ensure consistency across all baselines in terms of architectures, libraries, and other implementation details.
> >
> > [1] Santurkar S, Tsipras D, Ilyas A, Madry A. How does batch normalization help optimization?. Advances in neural information processing systems. 2018;31.
> > [2] Srinivasan R, Barausse E, Korsakova N, Trotta R. Simulation-based population inference of LISA's Galactic binaries: Bypassing the global fit. arXiv preprint arXiv:2506.22543. 2025 Jun 27.
> > [3] Lemos P, Parker L, Hahn C, Ho S, Eickenberg M, Hou J, Massara E, Modi C, Dizgah AM, Blancard BR, Spergel D. SimBIG: Field-level Simulation-Based Inference of Galaxy Clustering. arXiv preprint arXiv:2310.15256. 2023 Oct 23.
> > [4] Papamakarios G, Pavlakou T, Murray I. Masked autoregressive flow for density estimation. Advances in neural information processing systems. 2017;30.
> > [5] Wehenkel A, Louppe G. Unconstrained monotonic neural networks. Advances in neural information processing systems. 2019;32.

---

> > > ### Author Response · Authors · 2025-08-05
> > > **2/2 (SIR benchmark LPP table)**
> > >
> > > LPP scores calculated for the SIR benchmark ($\lambda=1, \gamma=0.5$)
> > > | Method/size of calib. set               | 10                  | 50                  | 200                 | 1000                |
> > > |----------------------|---------------------|---------------------|---------------------|---------------------|
> > > | SBI                  | 0.55913097                                                   |
> > > | NPE                  | 0.39261161                                                   |
> > > | OT-only (single sample) | -1.92932739                                                  |
> > > | OT-only (full test)  | 0.02674149                                                   |
> > > | Finetune-only        | -0.2342 ± 0.6621    | -0.0402 ± 0.2866    | 0.3441 ± 0.0560     | 0.3817 ± 0.0389     |
> > > | RoPE (single sample) | -1.8487 ± 0.2048    | -2.3768 ± 0.8745    | -1.9174 ± 0.0554    | -1.8415 ± 0.1035    |
> > > | RoPE (full test)     | -0.0227 ± 0.0594    | -0.0168 ± 0.0478    | -0.0065 ± 0.0216    | 0.0195 ± 0.0160     |
> > > | Ours                 | -0.2849 ± 0.2571    | -0.1904 ± 0.1937    | -0.0876 ± 0.0159    | -0.1001 ± 0.0150    |

---

> ### Comment · Reviewer_jspK · 2025-08-06
>
> I appreciate the additional experiments and clarifications. I don't have any more questions at this point, and I will raise my score to a 4.

---

> > ### Author Response · Authors · 2025-08-07
> >
> > Thank you for the productive discussion and for raising points that help strengthen and clarify our paper.

---

### Official Review · Reviewer_wUUr · 2025-06-28

**Clarity:** 3
**Significance:** 3
**Originality:** 3
**Rating:** 5
**Confidence:** 3

**Summary:**

This paper deals with the issue of misspecification in amortized simulation-based inference (SBI).
Specifically, in amortized SBI such as neural posterior estimation (NPE) one would train a neural network and possibly a neural statistics encoder such that given a dataset $x$ and encoded dataset $h(x)$ --> the network learns to map to the posterior $q(\theta| h(x))$.
The issue is that $h(x)$ and $q(\cdot)$ are trained using synthetic data (via a simulator). The real generative process may deviate from the simulator, leading to model misspecification / covariate shift.

Previous work (RoPE) addressed the issue via optimal transport, but had strong requirements (e.g., requiring a batch of test samples for inference). This work addresses multiple limitations of RoPE, including provided an amortized posterior estimation method based on optimal transport. The paper first introduces the method for joint supervised & OT training, and then how to amortize it.

The technique is evaluated on four benchmarks, with additional ablation studies and an analysis of robustness to noise.

**Questions:**

Please see above for my open comments and related questions.
I'd be happy to increase my score if the authors address my points.

**Ethical Concerns:**

["NO or VERY MINOR ethics concerns only"]

**Final Justification:**

The authors have addressed my concerns and after considering the whole discussion and rebuttal I increased my score.

**Limitations:**

The paper could be more upfront about its limitations -- naturally it inherits limitations of NPE, but it should be added which additional limitations are brought in by the specific methodology.

**Quality:**

3

**Strengths And Weaknesses:**

Overall, this is a solid paper tackling a very timely and significant issue, which is robustness of amortized SBI techniques to model misspecification (also known as *sim2real gap*). The approach is original and using state-of-the-art techniques, such as optimal transport (OT). The paper is generally well written and clear.

One potential weakness of the paper is that all considered baselines focus on OT approaches which require a "calibration dataset" (except for the NPE baseline) which consists of real data samples together with their true parameters. Obtaining such data for some applications may be a strong or costly requirement.
Given this additional constraint, it'd be useful for the practitioners to see how the proposed method compares to *robust* SBI techniques (beyond vanilla approaches) that do not require real data for calibration (see e.g. Huang et al., "Learning Robust Statistics for Simulation-based Inference under Model Misspecification", 2023). I would expect the paper's method to be better, since it leverages this additional information from the calibration set, but it'd be good to indeed see that this is the case.

As a minor exposition point, I recommend to state clearly the dimensions involved in each problem; that is parameter dimensionality (I think it's 2-4-1-4 or similar) and the dimensionality of the summary statistics. Relatedly, it'd be useful to know to which dimension (of parameters and data) is the method expected to scale.

---

> ### Author Rebuttal · Authors · 2025-07-31
>
> Thank you for the constructive and encouraging review, and for highlighting both the strengths of our approach and valuable directions for comparison and clarification. We address the reviewer’s comments and suggestions below.
>
> > W1. comparison with non-calibration set baselines
>
> This is indeed a very important point. In the ICML paper of the baseline RoPE work, the authors report the performance of two baselines without a calibration set: NPE-RS (Huang et al.) and NNPE (Ward et al., "Robust Neural Posterior Estimation and Statistical Model Criticism"). They show that while these methods perform better than vanilla NPE, they still fall short compared to baselines that incorporate a calibration set — even when the calibration set is small. We will add this to the discussion of the baseline experiment results (Section 4.3.1).
>
>
> > W2. As a minor exposition point, I recommend to state clearly the dimensions involved in each problem; that is parameter dimensionality (I think it's 2-4-1-4 or similar) and the dimensionality of the summary statistics. Relatedly, it'd be useful to know to which dimension (of parameters and data) is the method expected to scale.
>
> Thank you for the comment. We will clarify this point in the revised manuscript. In addition, we will include a discussion on scalability with respect to dimensionality. Indeed, we consider here only low-dimensional use cases, which are common in various real-world applications. However, the discussion regarding applicability to higher-dimensional settings is similar to that presented in the baseline RoPE work. The higher-dimensional parametric space identifiability problem could be solved by generating and training the NPE over more simulated samples.
>
> The main limitations in higher dimensions are twofold:
> 1. The misspecification between simulated and real interactions across the different $\theta_k$ could become more significant with the increase in dimension, increasing the complexity of the matching of embeddings of real observations with simulated ones. This requires larger sizes of calibration sets.
> 2. The embedding dimension required to encode sufficient statistics for predicting the posterior is expected to grow with the parameter dimensionality, which may complicate the coupling problem. In Chen et al. [1], they propose and validate a heuristic where the dimension of the summary statistics embedding is set to $d = 2K$, where $K$ is the dimension of $\theta$. In our implementation, we use an overparameterized embedding dimension of 16 regardless.
>
> [1] Chen Y, Zhang D, Gutmann M, Courville A, Zhu Z. Neural approximate sufficient statistics for implicit models. Proceedings of the International Conference on Learning Representations (ICLR), 2021.
>
> However, we expect our framework to alleviate these limitations. Specifically, the joint training of the OT and supervised objectives could help in situations where the small calibration set is insufficient to project the representation of the real observations onto the simulated ones. This can be balanced with the unsupervised OT coupling, which can be trained on an abundance of unlabelled data.
>
> Moreover, our choice of using $\rho=0$ benefits from a closed-form solution for the coupling, computed independently for each sample in $D_u$, involving $N_{OT}$ operations. Each operation consists of computing the Euclidean distance between a pair of embeddings in $\mathbb{R}^d$, which is achieved in $O(d)$. Since we operate in the mini-batch OT setting we have a complexity at each step of $B_t N_{OT} d$, where $B_t$ is the batch size. In comparison, RoPE which considers $\rho>0$ and cannot operate on mini-batches has a complexity of $O(log(N_u N_{OT})N_u N_{OT} d)$ where $N_u$ is the overall number of real observations. Hence, our approach would scale better than RoPE in these high-dimensional settings.

---

> > ### Comment · Reviewer_wUUr · 2025-08-01
> > **Response to authors**
> >
> > Thank you for your response and clarifications. I remain positive inclined towards acceptance, in preparation for the discussion with the other reviewers.

---

### Official Review · Reviewer_TtJD · 2025-07-02

**Clarity:** 3
**Significance:** 3
**Originality:** 2
**Rating:** 4
**Confidence:** 3

**Summary:**

The paper proposes an inductive and amortized method in the context of simulation based inference (SBI). Recent published works on SBI, RoPE operate in a transductive setting which uses optimal transport to align simulations to real data, but this needs recomputing the transport equation every for every inference time, which is a major limitation. This paper overcomes this limitation by using a joint training framework for the simulations and real data and an amortised posterior estimation. Experiments on a set of simulated and real world benchmarks provide evidence for the viability of this method.

**Questions:**

I mainly want some clarifications and some questions:

1) To make sure I understand correctly: the supervised calibration loss is mainly useful when the calibration pairs are accurate, while the OT loss provides robustness when those pairs are noisy or mismatched. Is that correct?
2) The evaluation datasets have only up to a maximum of four parameters. Oftentimes, complex simulations have a much larger parameter space in fields where SBIs are frequently used, such as complex fluids or astrophysics datasets. Can the authors describe how they expect their model to perform when the parameter dimensionality is very high?

**Ethical Concerns:**

["NO or VERY MINOR ethics concerns only"]

**Final Justification:**

I maintain my positive assessment for the paper given the authors have satisfactorily answered most of my questions. However, I do not think the paper will have that high of an impact to the field which is what the definition of a higher rating requires. But overall, I think this is a good paper.

**Limitations:**

Yes.

**Paper Formatting Concerns:**

No major issues found.

**Quality:**

3

**Strengths And Weaknesses:**

Strengths:
1) The paper is clearly written and most of the components going into the framework are explained reasonably well. Figures are clear as well.
2) Addresses an important problem of simulator misspecification in SBI. This is important to sim to real problems in general.
3) Clear motivation for improving over RoPE’s transductive limitations.
4) This method could be practically useful since it avoids simulation access at test time which is neat.
6) The paper provides useful ablation studies to show the importance of the joint training and amortization components.

Weaknesses:
1) An intuitive explanation of optimal transport is not provided in the main text. This can make it somewhat hard to understand for the readers.
2) The method seems to be suspectible to noise in the calibration set, which can potentially limit its applicability in a wide range of real world problems where calibration sets are scarce and noisy.
3) The paper uses mini-batch OT with a fixed entropy regularization parameter gamma, but does not explore how sensitive results are to this value.
4) The method is evaluated only on problems with up to four parameters. SBI applications often involve ten or more parameters. It is unclear how the OT coupling or the normalizing flow posterior will scale in such regimes, or whether performance degrades under the curse of dimensionality.

---

> ### Author Rebuttal · Authors · 2025-07-31
>
> Thank you for the constructive review and for highlighting the practical benefits of our approach and key areas for improvement. We address the reviewer’s concerns point by point below.
>
> ## Weaknesses:
>
> > W1. An intuitive explanation of optimal transport is not provided in the main text. This can make it somewhat hard to understand for the readers.
>
> Thank you for the suggestion, we will clarify the motivation and operations involved in Equation 1 in the corresponding section of the revised manuscript.
>
>
> > W2. The method seems to be suspectible to noise in the calibration set, which can potentially limit its applicability in a wide range of real world problems where calibration sets are scarce and noisy.
>
> It is true that we have identified some sensitivity to noise in the synthetic noise experiment. However, our approach worked better than the RoPE-single-sample baseline, and since RoPE-full-test is calculated and evaluated on the test set itself, it is perhaps able to couple even noisy samples when the noise model is relatively simple.
>
> It should be emphasized that in the real noisy benchmark (the cardiac biomarkers), our method outperformed RoPE. For clarity, we will emphasize this point in the revised manuscript in the results discussion.
>
>
>
> > W3. The paper uses mini-batch OT with a fixed entropy regularization parameter gamma, but does not explore how sensitive results are to this value.
>
> We performed a sensitivity analysis w.r.t the entropy regularisation param $\gamma$ on the light tunnel benchmark. The results below are for the lpp metric.
>
>
> | num_samples (calib.) ↓ / γ → | 0.01           | 0.1            | 0.5            | 1              | 5              |
> |---------------------|----------------|----------------|----------------|----------------|----------------|
> | 10                  | -8.83 ± 2.92   | -6.83 ± 2.19   | -2.57 ± 1.04   | -1.85 ± 0.69   | -2.27 ± 0.85   |
> | 50                  | -0.98 ± 0.60   | -0.54 ± 0.41   |  0.09 ± 0.28   |  0.02 ± 0.12   | -1.40 ± 0.06   |
> | 200                 |  0.13 ± 0.24   |  0.47 ± 0.17   |  0.66 ± 0.03   |  0.37 ± 0.04   | -1.27 ± 0.07   |
> | 1000                |  1.00 ± 0.08   |  1.15 ± 0.06   |  0.96 ± 0.05   |  0.54 ± 0.03   | -1.22 ± 0.06   |
>
> Like in RoPE, we observe for our method that when the calibration set is small, it is beneficial to select a larger value for the entropy regularization parameter, as the OT matching becomes more diffuse i.e closer to an uniform distribution. On the other hand, when the calibration set is large (e.g., 1000 samples), it is more likely that a good pairing exist between those and the real data, hence we can reduce $\gamma$ to obtain a sharper coupling.
>
>
>
>
> > W4. + Q2 The method is evaluated only on problems with up to four parameters. SBI applications often involve ten or more parameters. It is unclear how the OT coupling or the normalizing flow posterior will scale in such regimes, or whether performance degrades under the curse of dimensionality.
>
> Indeed, we consider here only low-dimensional use cases, which are common in various real-world applications. However, the discussion regarding applicability to higher-dimensional settings is similar to that presented in the baseline RoPE work. The higher-dimensional parametric space identifiability problem could be solved by generating and training the NPE over more simulated samples.
>
> The main limitations in higher dimensions are twofold:
> 1. The misspecification between simulated and real interactions across the different $\theta_k$ could become more significant with the increase in dimension, increasing the complexity of the matching of embeddings of real observations with simulated ones. This requires larger sizes of calibration sets.
> 2. The embedding dimension required to encode sufficient statistics for predicting the posterior is expected to grow with the parameter dimensionality, which may complicate the coupling problem. In Chen et al. [1], they propose and validate a heuristic where the dimension of the summary statistics embedding is set to $d = 2K$, where $K$ is the dimension of $\theta$. In our implementation, we use an overparameterized embedding dimension of 16 regardless.
>
> [1] Chen Y, Zhang D, Gutmann M, Courville A, Zhu Z. Neural approximate sufficient statistics for implicit models. Proceedings of the International Conference on Learning Representations (ICLR), 2021.
>
> However, we expect our framework to alleviate these limitations. Specifically, the joint training of the OT and supervised objectives could help in situations where the small calibration set is insufficient to project the representation of the real observations onto the simulated ones. This can be balanced with the unsupervised OT coupling, which can be trained on an abundance of unlabelled data.
>
> Moreover, our choice of using $\rho=0$ benefits from a closed-form solution for the coupling, computed independently for each sample in $D_u$, involving $N_{OT}$ operations. Each operation consists of computing the Euclidean distance between a pair of embeddings in $\mathbb{R}^d$, which is achieved in $O(d)$. Since we operate in the mini-batch OT setting we have a complexity at each step of $B_t N_{OT} d$, where $B_t$ is the batch size. In comparison, RoPE which considers $\rho>0$ and cannot operate on mini-batches has a complexity of $O(log(N_u N_{OT})N_u N_{OT} d)$ where $N_u$ is the overall number of real observations. Hence, our approach would scale better than RoPE in these high-dimensional settings.
>
> We notice that the points regarding dimensionality and scalability are repeatedly brought up by the reviewers, and we are planning to add a discussion of these aspects in the Conclusions and Limitations section.
>
>
>
> ## Questions
>
>
> > Q1. To make sure I understand correctly: the supervised calibration loss is mainly useful when the calibration pairs are accurate, while the OT loss provides robustness when those pairs are noisy or mismatched. Is that correct?
>
> Indeed, that is the case, but not only. Recall that the main setting we consider here is one where the calibration set is too small for a purely supervised domain transfer. To address this, we introduce an additional unsupervised OT loss. The OT loss provides a regularisation for matching real observations with simulated ones, even when the calibration set is small, which is the basic setting of the problem we aim to solve.
>
> This effect is evident from the baseline experiments in Figure 2, where we observe that our method outperforms the finetune-only baselines at small calibration set sizes. However, as the size of the calibration set increases, the performance gap diminishes.
>
>
> > Q2. dimensionality and scalability
>
> Please refer to W4

---

> > ### Comment · Reviewer_TtJD · 2025-08-02
> >
> > Thank you for the clarifications. I will retain my positive assessment of the paper given the authors have responded to my questions satisfactorily.

---

### Official Review · Reviewer_V4WH · 2025-07-04

**Clarity:** 3
**Significance:** 3
**Originality:** 3
**Rating:** 5
**Confidence:** 4

**Summary:**

This paper proposes an inductive and amortized framework for simulation-based inference
(SBI) under model misspecification, building upon the recent RoPE method. The key
innovation is combining optimal transport (OT)-based distribution alignment with
supervised calibration in a joint training objective, followed by amortization through a
conditional normalizing flow. This enables single-sample inference without requiring
simulations at test time, addressing RoPE's transductive limitations. The method is
evaluated on synthetic and real-world benchmarks, showing competitive or superior
performance to RoPE while offering practical deployment advantages.

**Questions:**

1. In Section 3.2, am I correct in understanding that the conditional normalizing flow
   learns an implicit mapping from real observations to parameters through the following
   transitive relationship: (a) \theta^{(j,k)} \sim
   q_{\psi^*}(\theta|h_{\omega}(x_s^j)) gives parameters that likely generated
   simulation $x_s^j$, (b) High $\alpha_{ij}$ indicates that real observation $x_r^i$ is
   similar to simulation $x_s^j$ in embedding space, (c) The flow $q_\xi$ thus learns
   that if $x_r^i \approx x_s^j$ (high $\alpha_{ij})$, then parameters that generated
   $x_s^j$ are also likely for $x_r^i$?

2. For the real hemodynamics data, how is the calibration dataset obtained? How are real
   $\theta$ values measured in practice?

3. How many samples were used to compute the LPP metric? This is important to avoid bias
   as noted in Lueckmann et al. 2021.

4. How does the method scale to higher-dimensional parameter spaces? The current
   evaluation is limited to relatively low-dimensional problems.

5. Have you explored approaches to mitigate the reduced posterior confidence (ACAUC < 0)
   observed with larger calibration sets?

**Ethical Concerns:**

["NO or VERY MINOR ethics concerns only"]

**Final Justification:**

I maintain my positive score as the authors have thoroughly addressed all my concerns, committing to clarify the calibration set requirements in the introduction, add sensitivity analyses for λ and γ, and improve exposition throughout. The core contribution of making RoPE inductive and amortized is valuable for practical deployment of misspecified SBI. While minor issues remain (missing CS/SIR experiments, implementation discrepancies), the authors' responsive and constructive engagement with reviewer feedback demonstrates their commitment to improving the paper.

**Limitations:**

The authors provide a thoughtful discussion of computational limitations in the
conclusion. To further strengthen the paper, consider also highlighting earlier in the
manuscript:

- The requirement for calibration data with true parameters, which distinguishes this
  setting from standard SBI where only observations are available
- Potential scalability considerations for high-dimensional parameter spaces
- The observed trade-off between calibration set size and posterior confidence

These clarifications would help readers better understand when this approach is most
applicable.

**Paper Formatting Concerns:**

- Typo in Figure 1 caption: "anchors to the OT matching"
- Typo around line 65: "w.r to"

**Quality:**

4

**Strengths And Weaknesses:**

### Strengths

**Quality:**

- Strong technical contribution that addresses real limitations of existing methods
- Thorough experimental evaluation including ablations to understand component
  contributions
- Competitive results across benchmarks while being more practically deployable
- Clear acknowledgment of method limitations (e.g., reduced confidence with larger
  calibration sets)

**Clarity:**

- Clear problem motivation and terminology for misspecified SBI scenarios
- Good concise explanation of the baseline RoPE method
- Well-structured training procedure box that clarifies the algorithm
- Appropriate use of same benchmarks as RoPE for direct comparison

**Significance:**

- Addresses important practical limitations of RoPE (transductive nature, need for
  test-time simulations)
- Enables deployment of misspecified SBI methods in real-world scenarios requiring
  single-sample inference
- Demonstrates effectiveness even with small training sets (1000 samples)

**Originality:**

- Novel combination of joint OT-supervised training with closed-form solutions
- Creative use of conditional normalizing flows to amortize OT-based posteriors
- Principled approach to making transductive methods inductive

### Weaknesses

**Clarity:**

- **Section 2.2 (lines 83-97)**: The entropic OT formulation introduces the mathematical
  framework quite directly, which may be challenging for readers less familiar with
  optimal transport. Consider adding a brief intuitive explanation before Equation 1,
  perhaps describing how OT finds correspondences between real and simulated
  observations in the embedding space to handle the domain shift.

- **Section 3.2**: The amortization step involves a (for me, see question below) subtle
  but elegant insight that could benefit from additional explanation. When sampling
  \theta^{(j,k)} \sim q\_{\psi^*}(\theta|h\_{\omega^*}(x_s^j)), it may not be
  immediately clear to readers why these parameters sampled from simulation-conditioned
  posteriors serve as valid training targets for real observations. It might help to
  explicitly describe the transitive relationship: since $\theta$ generates $x_s^j$, and
  $x_s^j$ is matched to $x_r^i$ through OT weights, these $\theta$ values are also
  plausible for $x_r^i$.

- **Figure 1**: While the figure shows the two-stage process, the visual flow and
  notation could be clearer. For instance, $\rho_f$ is not defined in the caption, and
  the complex relationships in the cNF training are understandably difficult to capture
  visually. Perhaps simplifying the notation or adding a legend for the different
  symbols and arrow types could help readers navigate the diagram.

- **Dataset notation**: The various datasets used throughout the paper
 ($D_{calib}$, $D_u$, $D_{OT}$, $D_{test}$)
  serve different purposes that may not be immediately clear on first reading. A brief
  table or list before the Training Procedure box summarizing what each dataset contains
  and its role in the method could improve clarity - though the Training Procedure box
  itself is already quite helpful in this regard.

**Quality:**

- **Scalability analysis**: The experiments focus on relatively low-dimensional
  parameter spaces (≤4D). While these benchmarks allow direct comparison with RoPE, it
  would be valuable to include some discussion of how the method might scale to higher
  dimensions, or to explicitly acknowledge this as an area for future investigation.

- **LPP metric details**: Following Lueckmann et al. 2021's recommendations, it would be
  helpful to specify the number of samples used to compute the LPP metric, as this can
  affect the reliability of the estimates.

- **Hemodynamics calibration data**: The paper mentions thermodilution for obtaining
  ground truth cardiac output values (line 179), but additional details about how these
  measurements correspond to the ABP segments would help readers understand the data
  collection process and implications w.r.t the limitations in other SBI use cases.

**Significance:**

- **Calibration data requirement**: While the need for paired data with known $\theta$
  is inherent to the problem setting, this requirement could be stated more prominently
  in the introduction (similar to the RoPE paper). This would help readers immediately
  understand that this approach targets a specific subset of SBI problems where some
  ground truth is available.

- **Incremental nature**: The contribution builds naturally on RoPE by adding
  amortization, which is a logical and useful extension. While incremental, it addresses
  genuine practical limitations that would prevent RoPE's deployment in many real-world
  scenarios.

---

> ### Author Rebuttal · Authors · 2025-07-31
>
> We thank the reviewer for the thoughtful analysis of our work and address next the main concerns:
>
> ## weaknesses
>
> ### Clarity:
>
> > W1. Section 2.2 (lines 83-97): Adding OT background.
>
> We will clarify in the revised version the introductive parts on OT as the reviewer suggests, specifically in the Entropic OT coupling paragraph section 2.2. (essentially explaining what we mean by soft matching)
>
> > W2. Section 3.2: Clarifying final posterior estimation and amortisation
>
> Perhaps the following explanation can make Section 3.2 more intuitive and clear. From Section 3.1 (referred to as "joint training only" in the ablation study on p.8), we obtain an estimated posterior density that is effectively a mixture of simulation-based posteriors, with mixture coefficients learned through the method in 3.1. What Section 3.2 introduces is a way to infer this density directly from the real observation $x_r$. The idea of using such a mixture as an approximation of the true posterior $p(\theta \mid x_r)$ is derived by prior work, RoPE (see Eq. (4)--(5), Section 3). The key insight from RoPE is that the OT plan approximates the joint distribution over $(x_r, x_s)$. By leveraging this approximate joint distribution, we can marginalize over $x_s$ (which becomes a discrete sum in our setting) to estimate the posterior [to be clarified in the paper Section 2.2, L97]
>
>
> > W3. Figure 1: Clarifying visualisation.
>
> Thank you for these suggestions that will be taken into account to improve Figure 1 in the revised manuscript. We will clarify our workflow by illustrating more clearly the inputs/outputs, which parts are optimized and the objectives they are trained on. Additionally, we will include a visualization of the baseline vanilla NPE (and NSE), which serves as a preliminary stage for our framework on top of the figure.
>
> > W4. Dataset notation.
>
>
> We thank the reviewer for the helpful suggestion, we will add brief sentences in the Training procedure to describe each considered dataset.
>
> ### Quality:
>
> > W5. + Q4: dimensionality and scalability analysis:
>
> Thank you for this interesting remark. Indeed, we consider here only low-dimensional use cases, which are common in various real-world applications. However, the discussion regarding applicability to higher-dimensional settings is similar to that presented in the baseline RoPE work. The higher-dimensional parametric space identifiability problem could be solved by generating and training the NPE over more simulated samples.
>
> The main limitations in higher dimensions are twofold:
> 1. The misspecification between simulated and real interactions across the different $\theta_k$ could become more significant with the increase in dimension, increasing the complexity of the matching of embeddings of real observations with simulated ones. This requires larger sizes of calibration sets.
> 2. The embedding dimension required to encode sufficient statistics for predicting the posterior is expected to grow with the parameter dimensionality, which may complicate the coupling problem. In Chen et al. [1], they propose and validate a heuristic where the dimension of the summary statistics embedding is set to $d = 2K$, where $K$ is the dimension of $\theta$. In our implementation, we use an overparameterized embedding dimension of 16 regardless.
>
> [1] Chen Y, Zhang D, Gutmann M, Courville A, Zhu Z. Neural approximate sufficient statistics for implicit models. Proceedings of the International Conference on Learning Representations (ICLR), 2021.
>
> However, we expect our framework to alleviate these limitations. Specifically, the joint training of the OT and supervised objectives could help in situations where the small calibration set is insufficient to project the representation of the real observations onto the simulated ones. This can be balanced with the unsupervised OT coupling, which can be trained on an abundance of unlabelled data.
>
> Moreover, our choice of using $\rho=0$ benefits from a closed-form solution for the coupling, computed independently for each sample in $D_u$, involving $N_{OT}$ operations. Each operation consists of computing the Euclidean distance between a pair of embeddings in $\mathbb{R}^d$, which is achieved in $O(d)$. Since we operate in the mini-batch OT setting we have a complexity at each step of $B_t N_{OT} d$, where $B_t$ is the batch size. In comparison, RoPE which considers $\rho>0$ and cannot operate on mini-batches has a complexity of $O(log(N_u N_{OT})N_u N_{OT} d)$ where $N_u$ is the overall number of real observations. Hence, our approach would scale better than RoPE in these high-dimensional settings.
>
> We notice that the points regarding dimensionality and scalability are repeatedly brought up by the reviewers, and we are planning to add a discussion of these aspects in the Conclusions and Limitations section.
>
> > W6. + Q3 LPP metric details: LPP computations and number of samples
>
> The number of test samples for the pendulum/causal-chamber benchmarks is mentioned in lines 226-227 (1000 for all, the same scale of test set size as in RoPE). Indeed for the pulse-waves benchmark it is not explicitly mentioned, but it's also 1000. Thank you for noticing, we will add it to the revised manuscript. As for the computation of LPP, there is no need to sample from the posterior as the normalising flows model models the density function of the posterior and we just derive from it the log density value of the ground truth parameter. For the calibration score, the acauc, we sample a 1000 samples from each posterior density.
>
> > W7. Hemodynamics calibration data: data collection & labelling process
>
> Indeed, the process of aligning the CO measurements obtained via the thermodilution method to the corresponding ABP segment is important for understanding the root cause of the noisy labels. Due to space constraints, we did not elaborate on this in the paper, but we will include a paragraph about it in the supplementary material in the revised manuscript.
>
>
> ### Significance:
>
> > W8. Calibration data requirement: being more explicit about this requirement.
> We mention the use of a calibration set and the semi-supervised settings in lines 37-39 in the intro when mentioning the prior work of RoPE. However we can make it clearer that we follow the same settings in our work.
>
> ### Questions
>
> > Q1. Section 3.2, amortisation clarification.
>
>  The intuition is correct. As explained in response to W2 RoPE has already showed that the real posterior can be approximated by the mixture of simulated posteriors obtained by the coupling. So indeed the $\alpha_{ij}$ correspond to the level of "match" between the parameters $\theta$ of a real observation and those of a simulated one. In 3.2 however, we amortise this mixture-based density using normalising flows.
> > Q2. For the real hemodynamics data, how is the calibration dataset obtained? How are real $\theta$ values measured in practice?
>
> The heart rate is measured from ECG and the cardiac output by the invasive thermodilution. Then the HR and CO measurments are aligned with the corresponding pulse-wave segment using the patient's electronical record, which can induce noisy missaligned pairs of $x$ and $\theta$. We can elborate about that in the revised manuscript in the supp. material.
>
> > Q3. How many samples were used to compute the LPP metric? This is important to avoid bias as noted in Lueckmann et al. 2021.
>
> Please refer to W6
>
> > Q4. How does the method scale to higher-dimensional parameter spaces? The current evaluation is limited to relatively low-dimensional problems.
> Please refer to W5
>
> > Q5. Have you explored approaches to mitigate the reduced posterior confidence (ACAUC < 0) observed with larger calibration sets?
>
> We believe that it is essentially a matter of hyperparameter tuning on ($\gamma$, $\lambda$), that we did not empirically explored. We envision solutions for future works considering for instance adaptive strategies for $\lambda$ given $\gamma$. One of them could be to merge both losses into a unified transport problem with additional constraints on the transport plan where $\lambda$ would become similar to a lagrangian multiplier.

---

> > ### Comment · Reviewer_V4WH · 2025-08-04
> >
> > Thank you for the detailed and thoughtful responses to the review comments. I appreciate the authors' willingness to address the clarity concerns and provide additional technical details.
> >
> > **Regarding the clarifications provided:**
> >
> > I'm satisfied with the proposed revisions for clarity (W1-W4), particularly the explanation of the transitive relationship in the amortization step and the promise to improve Figure 1. The additional details about the hemodynamics data collection process will be valuable.
> >
> > The scalability analysis (W5) is thorough and addresses my concerns.
> >
> > **Points that align with reviewer jspK's concerns:**
> >
> > 1. **Missing CS and SIR experiments**: I share reviewer jspK's concern here. Even if these tasks show limited misspecification, including them (perhaps in the appendix) would:
> >    - Demonstrate robustness when misspecification is minimal
> >    - Enable direct comparison with the full RoPE benchmark suite
> >    - Show that the method doesn't degrade performance in well-specified settings
> >
> >    The argument that "vanilla NPE already performs well" on these tasks seems to miss the point - a method addressing misspecification should work across the spectrum of misspecification levels.
> >
> > 2. **Implementation differences**: The large discrepancies in baseline results are concerning. Could the authors provide:
> >    - A more detailed explanation of why batch normalization has such a dramatic effect on LPP
> >    - A complete list of implementation differences from RoPE
> >    - Perhaps results both with and without batch normalization for transparency
> >
> > **Additional suggestions:**
> >
> > - Consider adding the λ sensitivity analysis table from the rebuttal to the appendix
> > - The calibration data requirement should be mentioned more prominently in the introduction, as originally suggested
> >
> > Overall, the paper makes a solid contribution by making RoPE inductive and amortized. With these clarifications and the inclusion of missing experiments, it would be strengthened in clarity.

---

> > > ### Author Response · Authors · 2025-08-05
> > >
> > > Thank you for the productive discussion. As previously agreed, we will include the clarification regarding the calibration set in the introduction. Additionally, we plan to incorporate the sensitivity analysis with respect to $\lambda$ and $\gamma$ in the paper (the exact location will depend on space constraints). We have also addressed reviewer jspK’s additional inquiries.

---

### Decision · Program_Chairs · 2025-09-17

**Decision:**

Accept (poster)

**Comment:**

The paper introduces an inductive and amortized framework for simulation-based inference that integrates calibration and optimal-transport-based alignment into a single trainable pipeline, enabling efficient inference without simulation access at test time and showing strong performance across synthetic and real-world benchmarks. Reviewers generally found the contribution technically solid, practically relevant, and an important extension of RoPE, though they noted that the contribution is somewhat incremental. Sentiment across reviewers is positive overall, with consensus that the method improves applicability of SBI but with concerns about clarity, scalability to higher dimensions, sensitivity to noisy calibration data, and missing benchmark experiments. To further strengthen the work, the authors should candidly and prominently discuss limitations, including the dependence on calibration data with known parameters, potential scalability issues in high-dimensional settings, and trade-offs in posterior confidence with larger calibration sets.